# Genomic and transcriptomic analyses of aortic stenosis enhance therapeutic target discovery and disease prediction

Aortic stenosis (AS) is a common valvular heart disease and has no pharmacological therapies. We performed a multi-ancestry genome-wide association meta-analysis of 86,864 AS cases among 2,853,408 individuals, discovering 241 autosomal independent risk loci and 3 X chromosome risk loci. We additionally performed sex-stratified and ancestry-stratified genome-wide association studies (GWASs), identifying an additional 5 sex-specific risk loci, 11 risk loci in European ancestry individuals and 1 risk locus in African ancestry individuals. We also performed a transcriptome-wide association study using expression quantitative trait loci from human aortic valves, discovering 54 new genes for which genetically predicted expression influences the risk of AS. We then generated a new polygenic risk score for AS. Finally, we performed gene silencing experiments targeting biologically relevant genes identified by our GWAS. Silencing of *CMKLR1* and *LTBP4* in human valvular interstitial cells substantially decreased mineralization, implicating a role for polyunsaturated fatty acids and transforming growth factor β signaling in AS.

Aortic stenosis (AS) is a common valvular heart abnormality with an estimated global prevalence of more than nine million individuals[1]. However, there are no effective pharmacological therapies for AS, with four-year all-cause mortality rates of up to 34% for moderate and 45% for severe untreated disease[2]. While surgical or percutaneous aortic valve (AV) replacement is effective and increasingly safe, many still succumb to AS or are ineligible for procedures. An improved understanding of AS biology is necessary to identify preventive strategies for this common condition.

The majority of AS diagnoses are due to calcific aortic valve disease (CAVD), a fibrocalcific pathology of the AV resulting in calcific AS[3]. In contrast to AS resulting from congenital AV abnormalities, such as bicuspid or unicuspid AVs, calcific AS typically occurs at an older age in individuals with trileaflet valves. Family-based studies of calcific AS estimate heritability as high as 49%[4] and several community-based studies demonstrate familial aggregation of calcific AS[5,6]. Multiple prior genome-wide association studies (GWASs) confirm that calcific AS is a polygenic trait[7,8]. Collectively, prior GWAS identified 49

unique risk loci in primarily European genetic ancestry populations[7–15]. Implicated mechanisms include lipid metabolism, inflammation, adiposity, calcification and cellular senescence[16]. The most robustly replicated genetic risk factor for AS is *LPA*, encoding lipoprotein(a) (Lp(a)), which is a highly heritable LDL-like macromolecule containing apolipoprotein(a). Motivated by the genetic associations, clinical trials are underway to test whether Lp(a) lowering slows the progression of AS[17] (NCT05646381). However, a substantial proportion of the genetic architecture and mechanisms for AS remains unexplained.

To more comprehensively investigate the genetic architecture of AS, we leveraged data from 30 studies to (1) perform a large, ethnically diverse GWAS and transcriptome-wide association study (TWAS) to date of AS, (2) perform the first X chromosome analysis of AS, (3) characterize sex-specific and ancestry-specific genetic risk factors for AS, (4) prioritize causal variants, genes and canonical pathways, (5) generate a new polygenic risk score (PRS) for AS and (6) perform in vitro functional characterization of prioritized candidate causal genes in human valvular interstitial cells.

✉e-mail: pnatarajan@mgh.harvard.edu; g.thanassoulis@gmail.com

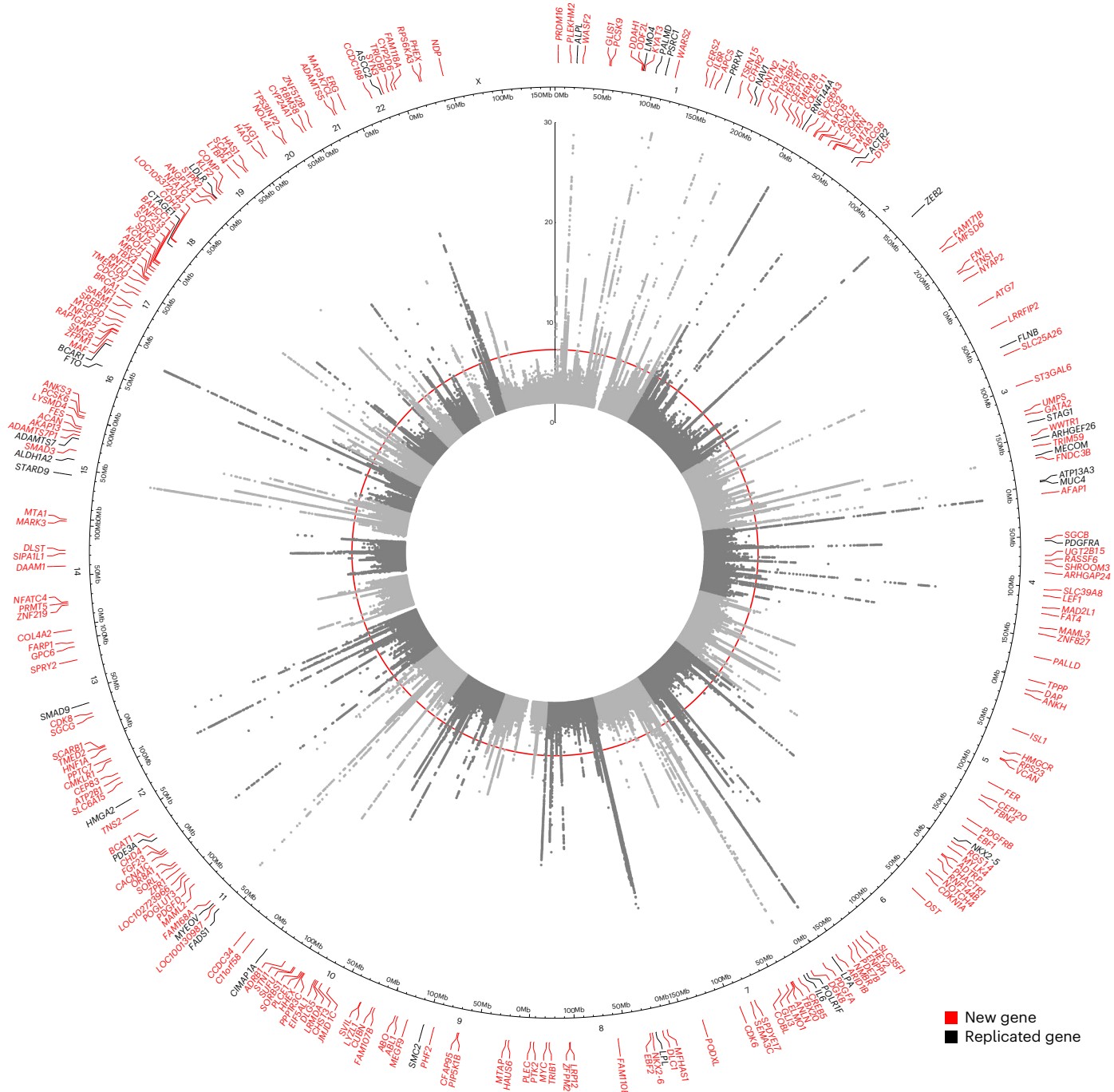

**Fig. 1 | Circos plot of multi-ancestry AS GWAS.** Circular Manhattan plot of our multi-ancestry AS GWAS results in both autosomes and the X chromosome. Genome-wide significance ($P = 5 \times 10^{-8}$), indicated by the inner red circle. Prioritized genes are highlighted at chromosomal position at outer circle, new loci are red and replicated loci are black.

## Results

### Genome-wide association analysis of AS

We performed a multi-ancestry GWAS of AS in autosomes in 2,853,408 individuals, representing 86,864 AS cases from 30 studies in the International Aortic Valve Genetics Consortium (IAVGC; Supplementary Fig. 1). Demographic characteristics by study are presented in Supplementary Table 1. The resulting GWAS meta-analysis included summary data on 54,133,673 variants present in at least two studies with a minimal minor allele count of ≥10 in each individual study. We identified 241 independent multi-ancestry lead variants, of which 38 were prior AS findings, and 203 were new (not in linkage disequilibrium (LD) ($r^2 < 0.8$) with any prior reported AS risk loci; Fig. 1, Supplementary Fig. 2 and Supplementary Table 3). A total of 187 (78%) multi-ancestry lead variants had a $P$ value for association with AS <$5 \times 10^{-9}$. We additionally performed a multi-ancestry association analysis of the X chromosome among 2,378,232 individuals, representing 69,877 AS cases. X chromosome analysis identified an additional three independent lead variants (Supplementary Fig. 3). Two lead variants from the multi-ancestry autosomal GWAS had significant heterogeneity in effect estimates between studies (rs74617384 and rs1474347); however, these remained genome-wide significant in a sensitivity analysis using random-effects inverse-variance-weighted

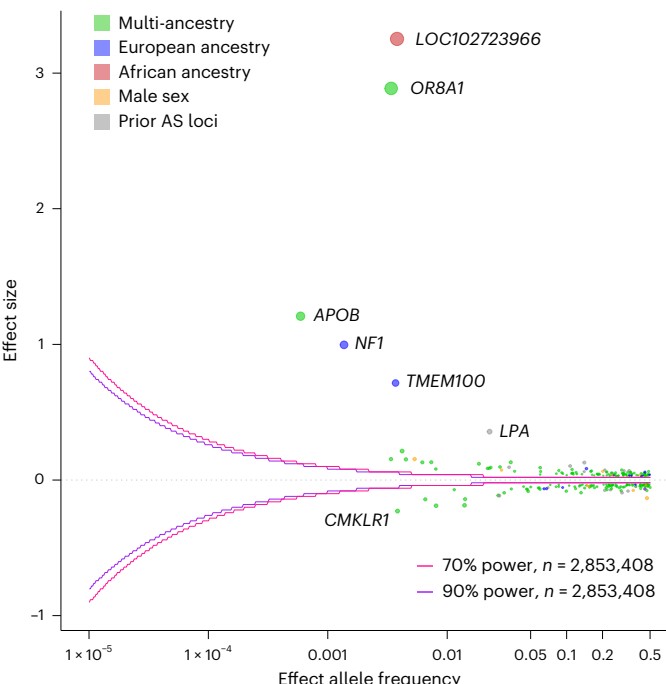

**Fig. 2 | Trumpet plot of lead variants in multi-ancestry as well as ancestry-stratified and sex-stratified AS GWASs.** Trumpet plot representing effect size and effect allele frequencies for lead variants. Lead variants are colored green if identified in multi-ancestry, gray if a prior AS risk locus, red if specific to African ancestry GWAS, blue if specific to European ancestry and yellow if specific to male sex GWAS. Variants with high effect sizes are labeled by the nearest gene.

meta-analysis (Supplementary Table 4). Of the 244 unique lead variants identified in multi-ancestry GWAS, 12 were rare (minor allele frequency (MAF) < 0.01, minimum MAF = 0.0006) and 20 were low frequency (MAF ≥ 0.01 and <0.05). The liability-scale heritability of AS in our multi-ancestry GWAS was 0.087 and the percent variance explained from lead variants was 5.5%.

When stratified by genetic ancestry, there were 2,464,395 individuals of European genetic ancestry (80,823 cases), 144,150 individuals of African genetic ancestry (3,126 cases), 66,444 individuals of Hispanic genetic ancestry (1,403 cases), 131,663 individuals of East Asian genetic ancestry (1,403 cases) and 43,950 individuals of South Asian genetic ancestry (109 cases). In the ancestry-stratified GWAS, we discovered 231 independent lead variants in European genetic ancestry and 2 independent lead variants in African genetic ancestry, but no genome-wide significant variants in Hispanic, East Asian or South Asian genetic ancestries (Supplementary Figs. 4–8 and Supplementary Table 5). Among the ancestry-stratified lead variants, 11 from the European genetic ancestry analysis and 1 from the African genetic ancestry analysis were independent from the 241 lead variants identified in the multi-ancestry meta-analysis. Of the 241 autosomal lead variants, effect estimates from the multi-ancestry analysis were most correlated with European ancestry GWAS results (Pearson's $r = 1.00$), followed by Hispanic ancestry ($r = 0.64$), African ancestry ($r = 0.34$), East Asian ancestry ($r = 0.14$) and South Asian ancestry ($r = 0.06$; Supplementary Fig. 9).

We performed sex-stratified analyses for 1,526,601 male individuals (55,795 cases) and 1,102,928 female individuals (22,639 cases). We identified 123 lead variants in male AS GWAS and 43 lead variants in female AS GWAS (Supplementary Figs. 10 and 11 and Supplementary Table 6). The vast majority of these were in LD with a previously identified multi-ancestry or ancestry-stratified GWAS; however, there were five male AS GWAS lead variants that were independent from lead variants discovered in our combined sex, multi-ancestry and ancestry-stratified GWAS (Supplementary Table 3). One of these

five sex-specific variants exhibited significant effect heterogeneity ($P < 0.05/5 = 0.01$) between males and females (rs17805623 (intronic to *RASSF6*); Supplementary Table 7). Of the 241 autosomal lead variants identified in the combined sex, multi-ancestry GWAS, 1 variant, rs7512646 (risk allele C frequency of 0.59), which is intronic to the *IL6R* gene, had significantly different effect estimates ($P_{heterogeneity} = 2.9 \times 10^{-6}$) between GWAS of males and females for AS (men−risk allele C, odds ratio (OR) = 1.03 (1.01–1.04), $P = 3.9 \times 10^{-5}$; women−risk allele C, OR = 1.09 (1.07–1.11), $P = 8.6 \times 10^{-16}$).

The resulting combined number of independent lead variants from all analyses totaled 261 (241 in multi-ancestry autosomal GWAS, 3 in multi-ancestry X chromosome analysis, 1 in African ancestry GWAS, 11 in European ancestry GWAS and 5 in male sex GWAS), of which 223 were new. Five of the 261 total risk loci had large effect estimates (risk allele OR for AS > 2). These were typically rare (MAF < 0.01), and most were noncoding with the exception of rs5742904 (*APOB* p.R3527Q), a known familial defective apolipoprotein B-100 pathogenic variant (Fig. 2). There were five variant pairs with an $r^2$ between 0.1 and 0.2 that were evaluated in conditional analysis (Supplementary Table 8). One variant, rs752446, which was identified in the female sex-stratified analysis, was not conditionally independent from other lead variants at the locus and was therefore not considered in our final count of independent lead variants.

## Transcriptome-wide association analysis

Genetic variation that modifies the risk of AS can be attributed to either systemic or AV-specific factors. To identify genes influencing AS risk through differential expression in the AV, we performed a TWAS using European ancestry AS GWAS data and human AV transcriptomes. A total of 192 genes reached a Bonferroni-corrected statistical significance threshold in TWAS, of which 66 showed significant colocalization between GWAS and expression quantitative trait loci (eQTLs; Supplementary Table 9 and Fig. 3). The majority of gene associations (82%, 54/66) were newly compared to prior published AS TWAS[9–11], among which 48 were at a genome-wide significant locus in our multi-ancestry or European GWAS. Eight genes had an expression specificity score (ESS) of >0.1, representing AV gene expression of more than 10% tissue-wide gene expression, including *COMP*, *LTBP2*, *ACAN*, *WNT9B*, *PDGFD*, *TMEM106A*, *PODXL* and *PRRX1*.

## Prioritization of candidate causal genes

We prioritized a single top gene for all 261 lead variants using a combination of annotation methods, including data from human AV tissue. Many prioritized genes were of high confidence with consensus by more than three methods (90/261, 34%; Fig. 4). Of the 261 prioritized genes, 127 (47%) were expressed in the AV as evidenced by having either a transcript or a protein product observed in proteomics or transcriptomic data of the human AV[18,19] or with genetically predicted expression substantially modifying risk of AS in TWAS using AV tissue transcriptomes. A total of 23 (9%) prioritized genes had an ESS >0.1. A total of 20 lead variants were in significant LD with a damaging, protein-coding variant (Supplementary Table 10). We additionally annotated the prioritized list of 261 genes by molecular function gene ontology using Enrichr. The top-most significant molecular function ontologies included transcription factor binding (*MYOCD*, *SMAD3*, *PRRX1*, *HMGA2*, *CHD4*), LDL particle binding (*SCARB1*, *PCSK9*, *SORL1*, *LDLR*) and growth factor receptor binding (*PDGFRB*, *PDGFRA*, *FER*, *IL6*), among other pathways (Supplementary Table 11). In contrast, the top-most significant molecular function ontologies for the 66 genes prioritized by AV TWAS included nuclear estrogen receptor binding (*NCOA6*, *LEF1*, *ISL1*), histone deacetylase binding (*LEF1*, *TWIST1*, *GLI3*) and transforming growth factor β (TGFβ) binding (*LTBP2*, *ITGAV*; Supplementary Table 12). The top cellular function ontology for the 66 genes prioritized by TWAS was actin cytoskeletal function (*SVIL*, *PALLD*, *TRIOBP*, *SIPA1L1*). Finally, we performed gene-set enrichment

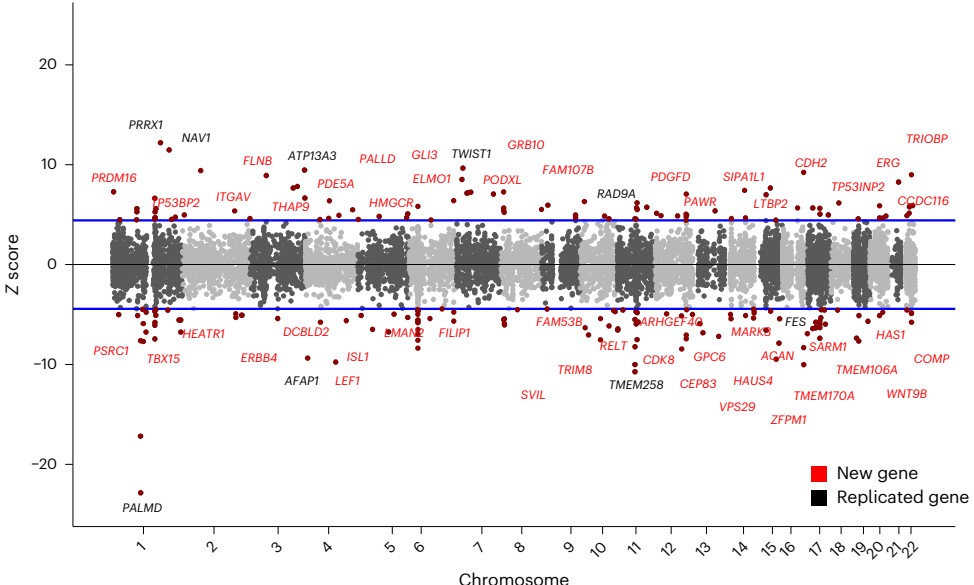

**Fig. 3 | Manhattan plot of TWAS using human AV eQTLs.** Manhattan plot of TWAS using European AS GWAS summary statistics and expression eQTLs data from human AVs. *Z* score represents magnitude and direction of association between genetically determined AV gene expression and AS risk. Genes in red are new TWAS associations. Only the most associated gene is labeled when multiple genes were identified at the same locus.

using DEPICT (Supplementary Fig. 12), which identified several additional pathways, including the Reactome pathways platelet aggregation (plug formation) and blood vessel development.

**Pleiotropic associations in AS**

We evaluated all 261 independent lead variants for pleiotropic associations across 2,068 traits, including 1,854 binary and 214 quantitative traits, using summary phenome-wide association study (PheWAS) data from the Million Veteran Program (MVP)[20]. A total of 220 (84%) lead variants had at least one pleiotropic association (Supplementary Table 13), among which 208 were multi-ancestry lead variants, 8 were European genetic ancestry lead variants and 3 were male sex-stratified lead variants. The strongest associations between continuous traits and risk of AS were among variants modifying lipid-relevant measures, including increased low-density lipoprotein (LDL) cholesterol (rs6511720 (prioritized to *LDLR*), rs11591155 (*PCSK9*), rs602633 (*PSRC1*)), lower high-density lipoprotein cholesterol (rs116843064 (*ANGPTL4*), rs115849089 (*LPL*)) and increased triglycerides (rs115849089 (*LPL*), rs780093 (*GCKR*)). Interestingly, the second strongest continuous trait association was between lead variants associated with increased risk for AS and increased height (rs2694594 (*GLIS1*), rs11161617 (*DDAH1*)). Pleiotropic associations for binary traits are presented in Supplementary Fig. 13. The top-most significant pleiotropic associations were between lead variants increasing risk for both AS and risk of disorders of lipid metabolism, occurring in lead variants prioritized for *PSRC1*, *PSCK9*, *LPA*, *TRIB1*, *ZPR1* and *LDLR*. Other notable pleiotropic associations for binary traits include the AS risk variant rs56094641 (*FTO*) and increased risk for both obesity and diabetes, and the AS risk variant rs780093 (*GCKR*) and increased risk of gout and cirrhosis.

**AS PRS**

We leveraged the large sample size of our GWAS to generate a new AS PRS to improve AS prediction. The best-performing score was derived using LDpred2 with a proportion of causal single-nucleotide polymorphisms (SNPs) of 0.056, heritability of 1.4 and with sparsity enabled. This score had a Nagelkerke $r^2$ for a model evaluating the PRS in prediction of AS in the Mass General Brigham Biobank (MGBB) of 0.007 (Supplementary Table 14). We validated this best PRS in the UKB (3,302 incident AS cases among 446,895 individuals), UCLA

ATLAS (1,806 incident AS cases among 34,024 individuals) and in aggregate data from six thrombolysis in myocardial infarction (TIMI) clinical trials (265 incident AS cases among 59,866 individuals). We also calculated an AS PRS using previously determined weights from a GWAS of AS among MVP individuals[21] and compared the efficacy of this older PRS to the present PRS in the UKB, TIMI clinical trials and UCLA ATLAS.

In all validation studies, the AS PRS outperformed the previously published PRS, with an overall twofold improvement in association (UKB−adjusted hazard ratio (HR) = 1.92 (1.85−1.99) per 1 s.d. PRS; TIMI trials−adjusted HR = 1.80 (1.60−2.04) per 1 s.d. PRS; Fig. 5a, Supplementary Table 15 and Supplementary Fig. 14). In a sensitivity analysis evaluating the association between the AS PRS and risk of AV replacement in the UKB (1,300 incident cases), the AS PRS was strongly associated with risk of AV replacement (HR = 2.00 (1.89−2.12)). In a multivariable model including clinical risk factors, high genetic risk (the top 20th percentile or above of the PRS compared to individuals with a PRS in the 40−60th percentile) had an estimated HR greater than all other risk factors, except the age of >65 years (Fig. 5b). Furthermore, there was a rising trajectory of risk with higher percentiles of the PRS. The C-index for clinical risk factors (age, sex, coronary artery disease (CAD), type 2 diabetes (T2D), hyperlipidemia (HLD), renal failure, current smoking, obesity) in the UKB was 0.85 (0.85−0.86), which improved to 0.87 (0.86−0.88) with the addition of the AS PRS (likelihood-ratio test *P* value for difference < 0.0001; Supplementary Fig. 15a). The C-index for clinical risk factors alone in TIMI trials was 0.71 (0.68−0.75), which improved to 0.76 (0.72−0.79) with the addition of the AS PRS (likelihood-ratio test *P* value for difference < 0.0001; Supplementary Fig. 15b). In both the UKB and TIMI clinical trials, the AS PRS had a C-index that was higher than all other individual risk factors except the age. Continuous net reclassification indices (NRIs) calculated between models with risk factors alone versus models with risk factors and the AS PRS were 29% (95% confidence interval = 20−37%) in the UKB and 23% (95% confidence interval = 15−30%) in the TIMI clinical trials. Finally, the AS PRS associated with AS in non-European populations in the UCLA ATLAS, with OR of 1.23 (1.01−1.49) for African ancestry, 1.38 (1.19−1.59) for admixed American ancestry and 1.38 (1.19−1.60) for East Asian ancestry. The AS PRS was also associated with AS among East Asian individuals in the UKB, with HR of 1.83 (1.41−2.40; Supplementary Table 15).

# Article

## Gene silencing and histopathology

We performed gene silencing experiments in human valve interstitial cells (VICs) from 11 donors for five genes (*CERS2*, *CEP120*, *LTBP4*, *CMKLR1*, *CLCA2*) that were prioritized using the following three criteria: (1) evidence for a direct pathogenic role in AS based on the genes harboring a genome-wide significant coding variant predicted to be damaging (Supplementary Table 11) and/or were the top prioritized gene in our causal gene prioritization pipeline (Supplementary Table 3), (2) evidence for AV-specific expression using data from prior human AV proteomics and/or transcriptomics datasets and (3) considered biologically relevant after review by content experts (S.I. and E. Aikawa). Transfection of siRNA targeting these five genes decreased expression in human VICs by between 57% (*LTBP4*) and 80% (*CLCA2*) relative to control siRNA-treated cells. Silencing of both *CMKLR1* and *LTBP4* significantly reduced calcification of VICs in osteogenic medium (*CMKLR1*, mean difference = −0.19 (staining intensity), *P* = 0.028; *LTBP4*, mean difference = −0.31 (staining intensity), *P* = 0.049; Fig. 6a,b). Silencing of *CLCA2*, *CERS2* or *CEP120*, however, did not substantially suppress calcification.

To validate our in vitro findings, we then performed immunofluorescence multilabeling of *LTBP4* and *CMKLR1* expression with osteogenic activity in human AVs (*n* = 3). Our results demonstrated localization of *LTBP4* in calcific nodules and muscle-like cells (Fig. 6c, left) and colocalization of *CMKLR1* with calcifying OsteoSense680-positive valvular cells and its expression in the AV microvessels (Fig. 6c, right, inset). Taken together, these observations support the notion that *LTBP4* and *CMKLR1* are highly expressed in calcifying human AVs.

## Discussion

We performed a GWAS of AS with 86,864 AS cases, which is four times the number of AS cases of previous studies, among a total sample of over 2.8 million individuals. We discovered 261 independent risk loci, replicating 38 prior AS risk loci, and contributing 223 (85%) new risk loci for AS. Among these, we identified five sex-specific AS risk loci, one African ancestry risk locus and three X chromosome risk loci. We prioritized potentially causal genes, proteins and pathways in AS and provided additional evidence for AV-specific effects by integrating human AV eQTL data. Finally, we generated a new AS PRS with approximately a twofold increase in association with AS per s.d. of the PRS, exceeding previously published scores.

Collectively, these data substantially advance our understanding of the genetic architecture of AS. We highlight several important insights. First, our data provide evidence for numerous new AV-specific genetic risk factors for CAVD. We found that many of the genes that were substantially associated with AS in TWAS using human AVs were implicated in actin cytoskeletal biology. For example, increased genetically predicted AV expression of *PALLD*, encoding the actin scaffolding protein Palladin, was associated with increased risk of AS. Palladin is thought to have important roles in myeloid differentiation and phagocytosis[22] and *PALLD* expression is increased in cardiac fibroblasts among individuals with restrictive cardiomyopathy[23]. *SVIL* encodes supervillin, an actin-binding protein with prior genetic evidence for associations with hypertrophic cardiomyopathy and descending aorta diameter[24,25]. Decreased genetically predicted AV expression of *SVIL* was associated with higher risk of AS in our study. Finally, *TRIOBP* encodes the TRIO and F-actin-binding protein that modulates the assembly

of the actin cytoskeleton[26]. TRIOBP is hypothesized to be involved in stress fiber formation and may have a pro-atherogenic phenotype[27]. We found that higher genetically predicted AV expression of *TRIOBP*

**Fig. 4 | Overview of causal gene prioritization.** Schematic depicting variants with a gene prioritized by at least three methods. Green represents multi-ancestry GWAS; blue represents European genetic ancestry. A total of 154 multi-ancestry unique lead variants, 8 European genetic unique ancestry lead variants, 5 male sex unique ancestry lead variants, 1 African genetic ancestry unique lead variant and 3 X chromosome unique lead variants with less than three supportive methods are not shown.

| Analysis | Variant | Prioritized gene | Number of supporting predictors |
|---|---|---|---|
| Multi-ancestry | rs6493981 | ALDH1A2 | 5 |
| | rs12430162 | CDK8 | 5 |
| | rs10417548 | HAS1 | 5 |
| | rs6511720 | LDLR | 5 |
| | rs115849089 | LPL | 5 |
| | rs7696431 | PALLD | 5 |
| | rs9674961 | RNF213 | 5 |
| | rs4990988 | TPPP | 5 |
| | rs1129448 | TRIOBP | 5 |
| | rs8176743 | ABO | 4 |
| | rs12594617 | ACAN | 4 |
| | rs62139061 | ACTR2 | 4 |
| | rs1801253 | ADRB1 | 4 |
| | rs62289340 | AFAP1 | 4 |
| | rs12141569 | ALPL | 4 |
| | rs116843064 | ANGPTL4 | 4 |
| | rs1706003 | ATP13A3 | 4 |
| | rs12458840 | CDH2 | 4 |
| | rs1639122 | CHD4 | 4 |
| | rs141421422 | CMKLR1 | 4 |
| | rs12974746 | COMP | 4 |
| | rs117870289 | ERG | 4 |
| | rs11130602 | FLNB | 4 |
| | rs4366594 | GPC6 | 4 |
| | rs2794763 | HEATR1 | 4 |
| | rs112009052 | LTBP4 | 4 |
| | rs3927738 | MTAP | 4 |
| | rs2293232 | MUC4 | 4 |
| | rs6702619 | PALMD | 4 |
| | rs4246336 | PCSK6 | 4 |
| | rs11591147 | PCSK9 | 4 |
| | rs75672964 | PODXL | 4 |
| | rs8006409 | PRMT5 | 4 |
| | rs602633 | PSRC1 | 4 |
| | rs11741640 | RGS14 | 4 |
| | rs113977592 | SDK2 | 4 |
| | rs13107325 | SLC39A8 | 4 |
| | rs2372785 | STRN | 4 |
| | rs1571759 | SVIL | 4 |
| | rs61741262 | TNS1 | 4 |
| | rs151058 | ADAMTS5 | 3 |
| | rs4243085 | ADAMTS7 | 3 |
| | rs1061813 | ANKH | 3 |
| | rs5742904 | APOB | 3 |
| | rs12710823 | ARHGAP24 | 3 |
| | rs57481061 | ATP2B1 | 3 |
| | rs12933281 | BCAR1 | 3 |
| | rs10219671 | BCAT1 | 3 |
| | rs10835169 | CCDC34 | 3 |
| | rs73372224 | CEP83 | 3 |
| | rs10483863 | DLST | 3 |
| | rs2064592 | DST | 3 |
| | rs6708393 | DYSF | 3 |
| | rs34370233 | ELMO1 | 3 |
| | rs453639 | ENPP1 | 3 |
| | rs36025028 | FAM107B | 3 |
| | rs6006988 | FAM118A | 3 |
| | rs36716 | FBN2 | 3 |
| | rs7637779 | FNDC3B | 3 |
| | rs1322756 | HEY2 | 3 |
| | rs1169288 | HNF1A | 3 |
| | rs1474347 | IL6 | 3 |
| | rs7512646 | IL6R | 3 |
| | rs7098614 | JMJD1C | 3 |
| | rs1840828 | KLF2 | 3 |
| | rs10863444 | LYPLAL1 | 3 |
| | rs34089221 | MEGF9 | 3 |
| | rs2465417 | MRC2 | 3 |
| | rs650720 | NAV1 | 3 |
| | rs2229309 | NFATC4 | 3 |
| | rs3131295 | NOTCH4 | 3 |
| | rs76709100 | ODF2L | 3 |
| | rs9801426 | PDGFA | 3 |
| | rs2019090 | PDGFD | 3 |
| | rs12523235 | PDGFRB | 3 |
| | rs1223580 | PLCE1 | 3 |
| | rs188595907 | PLEC | 3 |
| | rs715661 | RAP1GAP2 | 3 |
| | rs34161672 | RBM38 | 3 |
| | rs389411 | RNF144A | 3 |
| | rs389411 | RNF144A | 3 |
| | rs6948442 | SEMA3C | 3 |
| | rs556429 | SMAD9 | 3 |
| | rs17843768 | UMPS | 3 |
| | rs10060113 | VCAN | 3 |
| | rs36049560 | ZFPM1 | 3 |
| | rs964184 | ZPR1 | 3 |
| European | rs3803236 | COL4A2 | 4 |
| | rs2274432 | TSEN15 | 4 |
| | rs7149011 | ZNF219 | 3 |

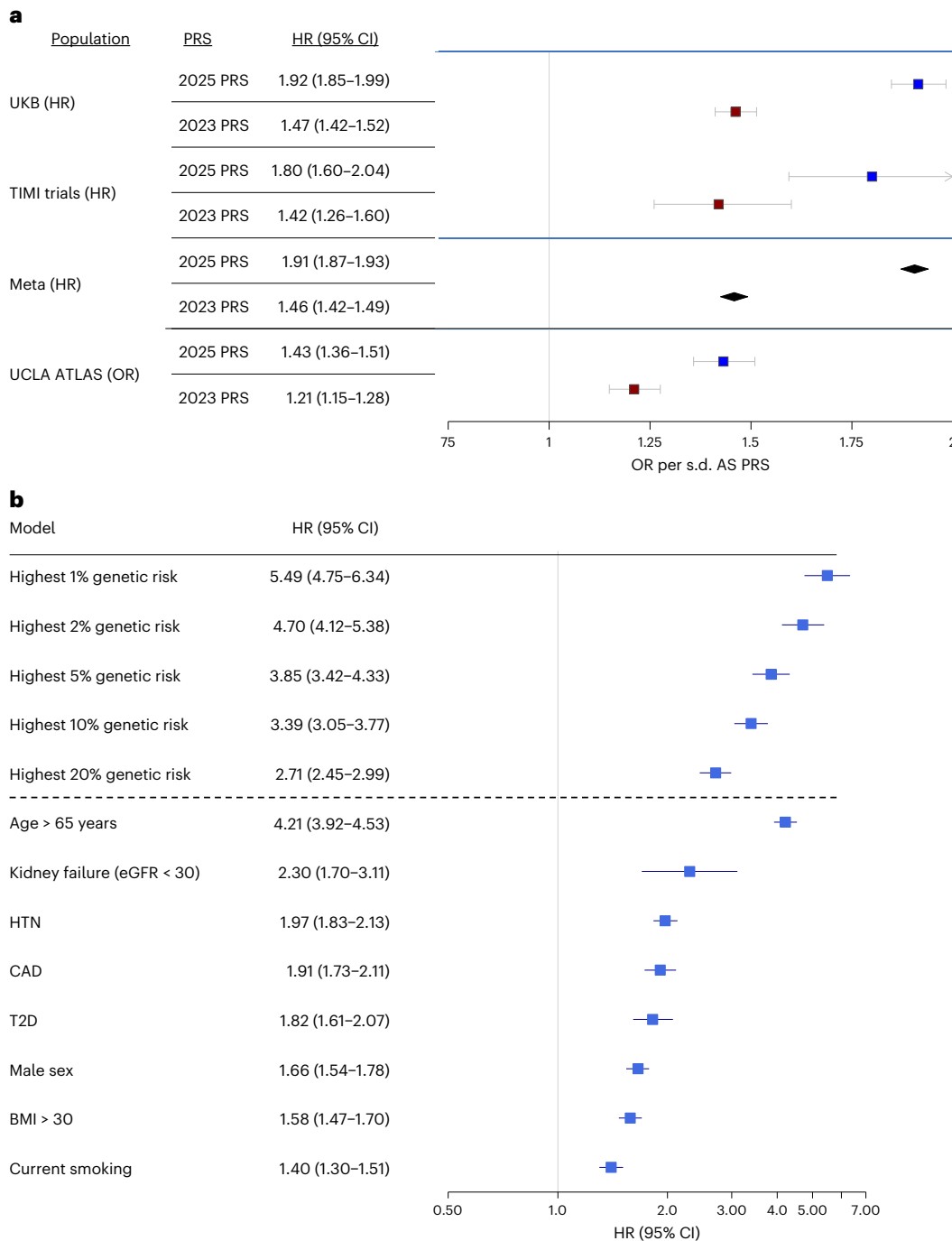

**Fig. 5 | Performance of AS PRS in European genetic ancestry individuals from the UK Biobank, TIMI trials and UCLA ATLAS. a**, Forest plot of AS PRS validation in the UK Biobank, TIMI clinical trials and UCLA ATLAS. **b**, Forest plot comparing the HR for categories of genetic risk and individual risk factors in the UKB. Forest plot in **a** demonstrating HRs/ORs and 95% CI per 1 s.d. PRS for the new AS PRS compared to a previously published PRS in the UK Biobank ($n$ = 446,895), TIMI trials ($n$ = 59,866) and the UCLA ATLAS ($n$ = 34,024), as well as a meta-analysis (Meta) of the UK Biobank and TIMI trial results. All analyses are adjusted for age, sex, five principal components and clinical risk factors including T2D, HTN, CAD, HLD, elevated BMI, current smoking and renal failure. Forest plot in **b** demonstrating HRs for genetic risk categories and individual risk factors with AS in the UKB ($n$ = 446,895). eGFR, estimated glomerular filtration rate; CI, confidence interval; BMI, body mass index.

was associated with increased risk of AS. While the actin cytoskeleton has diverse biological roles, several plausible mechanisms suggest that cytoskeletal biology may be particularly important in the development of AS. For example, activated human VICs may be differentiated into a pro-calcific phenotype through cytoskeletal-dependent mechanotransduction of AV flow perturbations[28].

In a previous GWAS[29], we described the importance of polyunsaturated fatty acid (PUFA) biosynthesis in the pathogenesis of AS by identifying genome-wide significant variation at the *FADS1/FADS2* locus, which we replicate in the current study. We extend evidence that PUFAs are risk factors for AS with a new GWAS finding of a genome-wide significant missense variant in *CMKLR1*, encoding chemerin chemokine-like receptor 1 (also known as ChemR23). ChemR23 is a G-protein coupled receptor for specialized pro-resolving lipid mediators (SPMs), which are downstream metabolites of PUFAs and function to resolve inflammation[30]. ChemR23 binds the SPM resolvin 1 (RvE1),

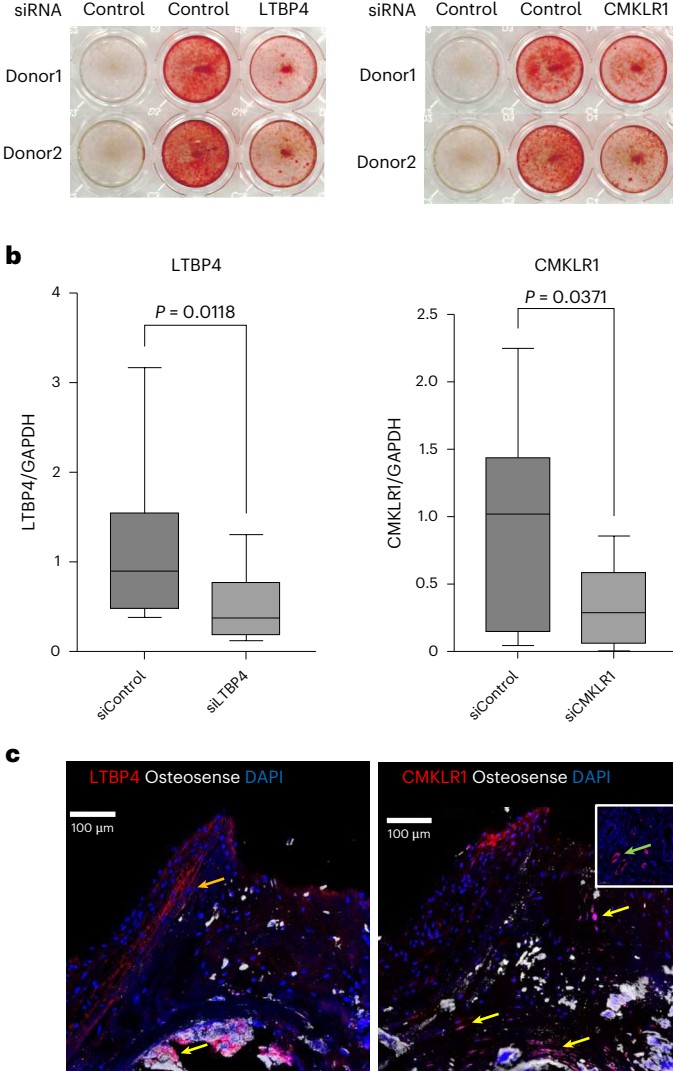

**Fig. 6 | Suppression of calcification in human VICs cultured in OM by silencing of LTBP4 and CMKLR1, and their colocalization with calcification in human AV. a**, Alizarin red staining identifies calcification in human VICs (example with two of eight donors). Silencing of LTBP4 and CMKLR1 significantly suppressed calcification in human VICs cultured in OM. Total $n = 11$ VIC donors. **b**, Validation of silencing of LTBP4 and CMKLR1 by siRNA transfection; qPCR. Silencing of LTBP4 and CMKLR1 significantly suppressed their gene expression levels in human VICs (mean ± s.d., $n = 8$; $P < 0.05$, two-tailed paired $t$ test). Box plots show the distribution from minimum to maximum values. The box represents the interquartile range (25th to 75th percentiles), with the line inside the box indicating the median. Whiskers extend to the minimum and maximum data points. **c**, Immunofluorescence staining of LTBP4 and CMKLR1 showed colocalization with calcific nodules and calcifying cells (OsteoSense680; yellow arrows) in human AV ($n = 3$ donors). LTBP4 expression in smooth muscle-like cells (orange arrow). CMKLR1 is colocalized with microvessels in human AV (green arrow, inset). Scale bars, 100 μm.

which is thought to be protective in both hypertension (HTN) and AV disease in mouse models[31,32]. Artiach and colleagues recently demonstrated that hyperlipidemic $Apoe^{-/-}$ mice with germline ChemR23 knockouts have higher AV gradients and that the addition of RvE1 to human VICs in culture suppresses calcification[31]. We identified a rare, missense variant in *CMKLR1* that has a substantial, protective effect against AS. We observed that silencing of *CMKLR1* in human VICs

suppressed calcification in osteogenic medium, a finding in apparent contrast to the proposed protective effect of this pathway. Although the exact reasons for these discordant results will require further study, there are potential explanations. First, ChemR23-deficiency may have different phenotypic consequences depending on the specific tissues involved, and germline knockouts may also activate compensatory regulatory mechanisms. Second, mouse models may not accurately mimic human biology for this pathway.

We also identified a rare, missense variant in *LTBP4* with a substantial, risk-enhancing effect for AS. We observed that silencing of LTBP4 in human VICs suppressed calcification in vitro. *LTBP4* encodes latent TGFβ binding protein 4, which functions both to bind and sequester free TGFβ[33], as well as to stabilize TGFβ receptors[34]. TGFβ has diverse and tissue-specific biologic roles; however, it is frequently implicated as a modulator of vascular calcification[35]. While it is still unclear how suppression of LTBP4 in valves directly impacts local TGFβ signaling, our data suggest that AV TGFβ signaling may be important in modifying pro-osteogenic pathways.

We identified sex-specific and ancestry-specific genetic risk factors for AS. There are several important sex-specific differences in the presentation of CAVD, most notable of which is the higher ratio of fibrosis to calcification in diseased AVs of females compared to males[36], without clear biologic explanation. We found one gene, *IL6R*, with a significant association with AS in females but not males, despite the male sample size being twice as large. *IL6R* encodes the interleukin-6 receptor, which is a well-established genome-wide significant variant in prior CAD GWAS and reached genome-wide significance in sex-stratified GWAS among women only suggesting differential sex effects due to inflammation. We found very few genome-wide significant variants in non-European populations, in large part due to limited sample sizes. The only independent genome-wide significant non-European genetic risk factor was observed in African genetic ancestry for the rare intronic variant rs148649124, found in *LOC102723966*, which is an uncharacterized long noncoding RNA, with as yet undetermined relevance in AV pathology. Notably, while there were no variants in the *LPA* region with genome-wide significance in any non-European populations, we did observe that the top-most significant *LPA* variant in prior AS GWAS, rs10455872, was subgenome-wide significant and directionally consistent in African ancestry individuals. There are several ongoing trials of Lp(a)-lowering medicines in clinical trials for atherosclerotic cardiovascular disease and some for AS (NCT05646381), but while robust human genetic evidence exists for Europeans, the evidence among non-Europeans remains sparse.

The majority of independent genome-wide-significant lead variants in AS had pleiotropic associations with other traits. Most notably, there were many strong associations between AS lead variants and lipid-relevant traits, including LDL cholesterol, high-density lipoprotein cholesterol, and triglycerides. This result is concordant with robust translational data suggesting that apolipoprotein B-containing lipoproteins, including Lp(a), are causal for AS[7,37,38]. Several clinical trials evaluating statin therapy in mild to moderate AS failed to demonstrate efficacy in lipid lowering as a strategy to mitigate adverse events in the disease[39–41]. However, these human genetic results highlight the potential role of much earlier lipid lowering for AS risk optimization.

Finally, we leveraged the size and diversity of our GWAS to generate a new AS PRS that had a twofold increase in risk association compared to a prior published AS risk score in multiple independent populations[21]. AS genetic risk was minimally attenuated by adjustment for common clinical risk factors and had an HR as high as, or higher than, all clinical risk factors except age. The addition of genetic risk to combined clinical risk factors modestly increased overall risk discrimination, with NRI values of 29% in the UKB and 23% in the TIMI clinical trials. We note that the C-index for clinical risk factors in predicting AS was higher in the UKB than in the TIMI clinical trials, which we attribute in

part to the UKB population being generally younger and healthier than those in the TIMI trials.

There are several limitations to the present work. To facilitate a uniform definition of AS across multiple diverse biobanks, we used a claims-based definition for AS which we previously validated in the MVP as having a positive predictive value of 0.78 and a negative predictive value of 0.99 (ref. 7). A key challenge in using claims data for diagnosis of AS is that ICD-9 coding for AV disease is limited to a single code (424.1) that captures both AS and aortic regurgitation. It is unclear whether mixed AV disease represents a unique pathobiology distinct from CAVD, but AS is far more common relative to aortic regurgitation. An additional limitation is that our population is predominantly of European genetic ancestry, limiting the generalizability of our findings. While the present study represents a substantial increase in sample size of non-European individuals, such individuals remain understudied in GWASs at large. Finally, the largest single contributor to both samples and cases in this study was the MVP, a predominantly (>95%) male population. The inclusion of the MVP skewed the distribution of cases to be majority male, which is not representative of the epidemiology of CAVD, where there is a more balanced prevalence of disease between men and women. Accordingly, we had better power for genetic discovery among men than women. Notably, bicuspid AVs are more prevalent in men than women, and while we attempted to explicitly exclude these cases, it is likely that a small number of AS cases in our GWAS were bicuspid.

In summary, we conducted a large GWAS of AS, identifying 261 independent risk loci, 225 of which were new. We describe new biologic insights in the pathobiology of AS, highlighting the relevance of several cytoskeletal genes and the importance of PUFA signaling via *CMKLR1*. Finally, we developed a new AS PRS with substantially improved prediction metrics compared to prior published AS PRS, as well as established individual clinical risk factors in multiple independent datasets.

## Online content

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

**Aeron M. Small**[1,2,3], **Ta-Yu Yang**[4,5,6,78], **Shinsuke Itoh**[1,7,78], **Sébastien Thériault**[8,9,78], **Line Dufresne**[4,78], **Ryo Kurosawa**[10], **Issei Komuro**[11,12], **Koichi Matsuda**[13], **Ha My T. Vy**[14], **Eric H. Farber-Eger**[15], **Lauren Lee Shaffer**[15], **Kristin M. Boulier**[16], **Kristin M. Corey**[17,18], **Megan E. Ramaker**[17], **Fabien Laporte**[19], **Jean-Jacques Schott**[19], **Solena Le Scouarnec**[19], **Sasha A. Singh**[1,7], **Abhijeet R. Sonawane**[1,7], **Harry A. Smith**[20], **Nicholas Rafaels**[20], **Colorado Center for Personalized Medicine***, **Jonas Ghouse**[21], **Anna A. Raja**[21], **Sisse R. Ostrowski**[22,23], **Erik Sørensen**[22], **Christina Mikkelsen**[22], **Ole B. Pedersen**[23,24], **Christian Erikstrup**[25], **Henrik Ullum**[26], **DBDS Genomic Consortium***, **Gardar Sveinbjornsson**[27], **Daniel F. Gudbjartsson**[27], **Erik Abner**[28], **Estonian Biobank Research Team***, **Jiwoo Lee**[2,29,30], **Andrea Ganna**[31,32], **Ulrike Nowak-Göttl**[33], **Sarah Finer**[34], **Genes & Health Research Team***, **Johannes Schumacher**[35,36], **Carlo Maj**[35], **Baravan Al-Kassou**[37], **Georg Nickenig**[37], **Teresa Trenkwalder**[38,39], **Martina Dreßen**[40], **Markus Krane**[39,40,41], **Markus M. Nöthen**[36], **Marta R. Moksnes**[42], **Ben M. Brumpton**[42,43], **Stacey Knight**[44], **Kirk U. Knowlton**[44], **Lincoln Nadauld**[44], **Radek Debiec**[45,46], **Muntaser D. Musameh**[45,46], **Peter S. Braund**[45,46], **Christopher P. Nelson**[45,46], **Tomasz Czuba**[47], **Olle Melander**[48], **Margaret Sunitha Selvaraj**[49], **Satoshi Koyama**[2], **Rohan Bhukar**[2], **Yunfeng Ruan**[2], **Johan Ljungberg**[50], **Scott M. Damrauer**[51,52], **Michael G. Levin**[53], **Andre Franke**[54], **Klaus Berger**[55], **Christian T. Ruff**[56], **Giorgio E. M. Melloni**[56], **Frederick K. Kamanu**[56], **Kaoru Ito**[10,57], **Ron Do**[14], **Ruth J. F. Loos**[14,58], **Heribert Schunkert**[38,39], **Quinn S. Wells**[15], **Svati H. Shah**[17,59,60], **Thierry Le Tourneau**[19], **David Messika-Zeitoun**[61], **Christopher Gignoux**[20], **Henning Bundgaard**[21], **Susanna C. Larsson**[62,63], **Karl Michaëlsson**[62], **Hilma Holm**[27], **Anna Helgadottir**[27], **Tonu Esko**[28], **David A. van Heel**[64], **Patrick Mathieu**[8,65], **Nilesh J. Samani**[45,46], **J. Gustav Smith**[47,66,67], **Stefan Söderberg**[50], **Daniel J. Rader**[53], **Nicholas A. Marston**[56], **Marc S. Sabatine**[56], **Bogdan Pasaniuc**[68], **Kelly Cho**[69,70,71], **Peter W. F. Wilson**[72], **Christopher J. O'Donnell**[1,3], **Kari Stefansson**[27], **Yohan Bossé**[8,73,79], **Elena Aikawa**[1,7,79], **James C. Engert**[4,5,79], **Gina M. Peloso**[74,79], **Pradeep Natarajan**[75,76,77,79] ✉ **& George Thanassoulis**[4,5,79] ✉

[1]Division of Cardiovascular Medicine, Brigham and Women's Hospital, Harvard Medical School, Boston, MA, USA. [2]Program in Medical and Population Genetics and the Cardiovascular Disease Initiative, Broad Institute of MIT and Harvard, Cambridge, MA, USA. [3]Division of Cardiology, Department of Medicine, VA Boston Healthcare System, Boston, MA, USA. [4]McGill University Health Centre and Research Institute, Montréal, Quebec, Canada. [5]McGill University, Montréal, Quebec, Canada. [6]Kyoto-McGill International Collaborative School in Genomic Medicine, Graduate School of Medicine, Kyoto University, Kyoto, Japan. [7]Center for Interdisciplinary Cardiovascular Sciences, Brigham and Women's Hospital, Harvard Medical School, Boston, MA, USA. [8]Institut Universitaire de Cardiologie et de Pneumologie de Québec-Université Laval, Quebec City, Quebec, Canada. [9]Department of Molecular Biology, Medical Biochemistry and Pathology, Université Laval, Quebec City, Quebec, Canada. [10]Laboratory for Cardiovascular Genomics and Informatics, RIKEN Center for Integrative Medical Sciences, Yokohama, Japan. [11]International University of Health and Welfare, Tokyo, Japan. [12]Department of Frontier Cardiovascular Science, Graduate School of Medicine, The University of Tokyo, Tokyo, Japan. [13]Institute of Medical Science, Laboratory of Genome Technology, Human Genome Center, The University of Tokyo, Tokyo, Japan. [14]The Charles Bronfman Institute for Personalized Medicine, Icahn School of Medicine at Mount Sinai, New York, NY, USA. [15]Department of Medicine, Division of Cardiology, Vanderbilt University Medical Center, Nashville, TN, USA. [16]Division of Cardiology, Department of Medicine, Ronald Reagan UCLA Medical Center, Los Angeles, CA, USA. [17]Duke Molecular Physiology Institute, Duke University, Durham, NC, USA. [18]Department of Medicine, Duke University, Durham, NC, USA. [19]Nantes Université, CHU Nantes, CNRS,

INSERM, l'institut du Thorax, Nantes, France. [20]Colorado Center for Personalized Medicine, University of Colorado Anschutz Medical Campus, Aurora, CO, USA. [21]Department of Cardiology, The Heart Centre, Copenhagen University Hospital, Rigshospitalet, Copenhagen, Denmark. [22]Department of Clinical Immunology, Copenhagen University Hospital, Rigshospitalet, Copenhagen, Denmark. [23]Department of Clinical Medicine, Faculty of Health and Medical Sciences, University of Copenhagen, Copenhagen, Denmark. [24]Department of Clinical Immunology, Zealand University Hospital, Køge, Denmark. [25]Department of Clinical Immunology, Aarhus University Hospital, Aarhus, Denmark. [26]Statens Serum Institut, Copenhagen, Denmark. [27]deCODE genetics/Amgen, Inc., Reykjavik, Iceland. [28]Estonian Genome Center, Institute of Genomics, University of Tartu, Tartu, Estonia. [29]Harvard-MIT Health Sciences and Technology Program, Harvard Medical School, Boston, MA, USA. [30]Cardiovascular Research Center and Center for Genomic Medicine, Massachusetts General Hospital, Boston, MA, USA. [31]Institute for Molecular Medicine Finland, HiLIFE, University of Helsinki, Helsinki, Finland. [32]Analytic and Translational Genetics Unit, Boston, MA, USA. [33]Thrombosis and Hemostasis Unit, Institute of Clinical Chemistry, University Hospital Kiel, Kiel, Germany. [34]Wolfson Institute of Population Health, Queen Mary University of London, London, UK. [35]Institute of Human Genetics, Philipps University of Marburg, Marburg, Germany. [36]Institute of Human Genetics, University of Bonn and University Hospital Bonn, Bonn, Germany. [37]Department of Medicine II, Heart Center Bonn, University of Bonn and University Hospital Bonn, Bonn, Germany. [38]Department of Cardiology, German Heart Centre Munich, TUM University Hospital, School of Medicine and Health, Technical University of Munich, Munich, Germany. [39]DZHK (German Center for Cardiovascular Research), Partner Site Munich Heart Alliance, Munich, Germany. [40]Department of Cardiovascular Surgery, Institute Insure, German Heart Center Munich, School of Medicine & Health, TUM University Hospital, Technical University of Munich, Munich, Germany. [41]Department of Surgery, Yale University School of Medicine, New Haven, CT, USA. [42]HUNT Center for Molecular and Clinical Epidemiology, Department of Public Health and Nursing, NTNU Norwegian University of Science and Technology, Trondheim, Norway. [43]Clinic of Medicine, St. Olavs Hospital, Trondheim University Hospital, Trondheim, Norway. [44]Intermountain Healthcare, Salt Lake City, UT, USA. [45]Department of Cardiovascular Sciences, University of Leicester, Leicester, UK. [46]NIHR Leicester Biomedical Research Centre, Glenfield Hospital, Leicester, UK. [47]The Wallenberg Laboratory/Department of Molecular and Clinical Medicine, Institute of Medicine and Science for Life Laboratory, Gothenburg University and Department of Cardiology, Sahlgrenska University Hospital, Gothenburg, Sweden. [48]Department of Internal Medicine, Clinical Sciences, Lund University and Skåne University Hospital, Malmö, Sweden. [49]Cardiovascular Research Center, Massachusetts General Hospital, Boston, MA, USA. [50]Department of Public Health and Clinical Medicine, Umeå University, Umeå, Sweden. [51]Department of Surgery, Perelman School of Medicine, University of Pennsylvania, Philadelphia, PA, USA. [52]Corporal Michael Crescenz, VA Medical Center, Philadelphia, PA, USA. [53]Division of Cardiovascular Medicine, Department of Medicine, University of Pennsylvania Perelman School of Medicine, Philadelphia, PA, USA. [54]Institute of Clinical Molecular Biology, Christian-Albrechts-University of Kiel, Kiel, Germany. [55]Institute of Epidemiology and Social Medicine, University of Münster, Munster, Germany. [56]TIMI Study Group, Brigham and Women's Hospital, Harvard Medical School, Boston, MA, USA. [57]Department of Advanced Biomedical Data Science, Graduate School of Medicine, Chiba University, Chiba, Japan. [58]Novo Nordisk Foundation Center for Basic Metabolic Research, Faculty of Health and Medical Science, University of Copenhagen, Copenhagen, Denmark. [59]Department of Medicine, Division of Cardiology, Duke University, Durham, NC, USA. [60]Duke Clinical Research Institute, Duke University, Durham, NC, USA. [61]Division of Cardiology, University of Ottawa Heart Institute, Ottawa, Ontario, Canada. [62]Medical Epidemiology, Department of Surgical Sciences, Uppsala University, Uppsala, Sweden. [63]Unit of Cardiovascular and Nutritional Epidemiology, Institute of Environmental Medicine, Karolinska Institutet, Stockholm, Sweden. [64]Blizard Institute, Queen Mary University of London, London, UK. [65]Department of Surgery, Université Laval, Quebec City, Quebec, Canada. [66]Department of Cardiology, Clinical Sciences, Lund University and Skåne University Hospital, Lund, Sweden. [67]Wallenberg Center for Molecular Medicine and Lund University Diabetes Center, Lund University, Lund, Sweden. [68]Interdepartmental Program in Bioinformatics, University of California, Los Angeles, CA, USA. [69]Department of Medicine, Harvard Medical School, Boston, MA, USA. [70]MVP Boston Coordinating Center, VA Boston Healthcare System, Boston, MA, USA. [71]Department of Medicine, Division of Aging, Brigham and Women's Hospital, Boston, MA, USA. [72]Atlanta VA Medical Center, Atlanta, GA, USA. [73]Department of Molecular Medicine, Université Laval, Quebec City, Quebec, Canada. [74]Department of Biostatistics, Boston University School of Public Health, Boston, MA, USA. [75]Cardiology Division, Massachusetts General Hospital, Harvard Medical School, Boston, MA, USA. [76]Broad Institute of MIT and Harvard, Cambridge, MA, USA. [77]Center for Genomic Medicine, Department of Medicine, Massachusetts General Hospital, Boston, MA, USA. [78]These authors contributed equally: Ta-Yu Yang, Shinsuke Itoh, Sébastien Thériault, Line Dufresne. [79]These authors jointly supervised this work: Yohan Bossé, Elena Aikawa, James C. Engert, Gina M. Peloso, Pradeep Natarajan, George Thanassoulis. *Lists of authors and their affiliations appear at the end of the paper. ✉e-mail: pnatarajan@mgh.harvard.edu; g.thanassoulis@gmail.com

## Colorado Center for Personalized Medicine

**Nicholas Rafaels[20] & Christopher Gignoux[20]**

## DBDS Genomic Consortium

**Sisse R. Ostrowski[22,23], Erik Sørensen[22], Christina Mikkelsen[22], Ole B. Pedersen[23,24], Christian Erikstrup[25] & Henrik Ullum[26]**

## Estonian Biobank Research Team

**Tonu Esko[28]**

## Genes & Health Research Team

**Sarah Finer[34] & David A. van Heel[64]**

## Methods

All participants for all studies provided written or verbal consent and studies were approved by the local ethics committee or institutional review board (IRB).

For the BioME study, protocols were approved by the IRB at the Icahn School of Medicine at Mount Sinai (GCO 07–0529; STUDY-11–01139) and all participants provided informed consent. For the BioVU study, all DNA samples in BioVU are de-identified and have been designated with the IRB, thus allowing the use of blood samples collected for clinical care otherwise scheduled for discard. The program has received IRB approval and was reviewed in detail by the federal Office for Human Research Protections, which agreed with the regulatory designation of the nonhuman participants. For the CathGen study, all participants provided informed consent, and the study was approved by the Duke University IRB.

CARTaGENE obtained ethics approval from the Centre Hospitalier Universitaire Sainte-Justine (reference MP-21-2011-345, 3297). The Danish analyses for CHB/DBDS were conducted within the CHB–CVDC and DBDS cohorts, which were approved by the Danish National Committee on Health Research Ethics (approval NVK-1708829 and NVK-1700407) and the Capital Region Data Protection Agency (approval P-2019-93 and P-2019-99). Participants in FinnGen provided informed consent for biobank research under the Finnish Biobank Act. Alternatively, separate research cohorts, collected before the Finnish Biobank Act came into effect (September 2013) and the start of FinnGen (August 2017), were collected on the basis of study-specific consent and later transferred to the Finnish biobanks after approval by Fimea, the National Supervisory Authority for Welfare and Health. Recruitment protocols followed the biobank protocols approved by Fimea. The Coordinating Ethics Committee of the Hospital District of Helsinki and Uusimaa (HUS) approved the FinnGen study protocol (HUS/990/2017).

The GERA study (Kaiser Permanente Research Program on Genes, Environment, and Health, RPGEH) was approved by the Grand Opportunity Project (IRB CN-09CScha-06-H). For Genes & Health, a favorable ethical opinion for the main genes and health research study was granted by NRES Committee London–South East (reference 14/LO/1240) on 16 September 2014. Queen Mary University of London is the sponsor and data controller. The analyses in HUNT have been approved by the Norwegian Data Protection Authority and the Regional Committee for Medical and Health Research Ethics (REC reference 2014/144).

Ethical approval for the Malmö Diet and Cancer study was obtained from the Lund University IRB, and all participants provided written informed consent. We acknowledge the Penn Medicine Biobank (PMBB) for providing data and thank the patient-participants of Penn Medicine who consented to participate in this research program. We also thank the PMBB team and Regeneron Genetics Center for providing genetic variant data for analysis. The PMBB is approved under IRB (protocol 813913) and supported by the Perelman School of Medicine at the University of Pennsylvania, a gift from the Smilow family, and the National Center for Advancing Translational Sciences of the National Institutes of Health under CTSA award UL1TR001878.

The Northern Swedish Health and Disease Study was approved by the Regional Ethical Review Board in Umeå (Dnr. 07-174 M, Dnr. 2014-348-32 M and Dnr. 2015-326-32 M). The analyses based on data from COSMC, SIMPLER and SMCC were approved by the Swedish Ethical Review Authority (Dnr. 2019-03986). The IUCPQ-UL study was approved by the ethics committee of IUCPQ-UL, and all participants provided written informed consent. For All of Us (AoU), written informed consent was provided in accordance with the primary IRB. AoU data analysis was facilitated through the AoU Researcher Workbench.

The Biobank Japan study was approved by the ethics committees of the RIKEN Center for Integrative Medical Sciences, the Institute of Medical Sciences and the University of Tokyo. Informed consent was obtained from all participants, all of whom were Japanese and registered in the BBJ project. The CAVS-France study was approved by the local ethics committees (CCPPRB Nantes, 404/2002; CPP Sud Méditerranée, 13.061; CCPPRB Hôtel-Dieu Paris, 0611285 and CPP Ile de France 1, 2014-juillet-13625) and all participants provided informed consent for genetic research.

The use of data from Iceland was approved by the National Bioethics Committee (NBC, VSN-15-057). All genotyped participants signed a written informed consent allowing the use of their samples and data in projects at deCODE genetics, approved by the NBC. The activities of the Estonian Biobank are regulated by the Human Genes Research Act, adopted in 2000 specifically for the Estonian Biobank. Individual-level data analysis in the Estonian Biobank was carried out under ethical approval 1.1-12/624 from the Estonian Committee on Bioethics and Human Research (Estonian Ministry of Social Affairs), using data according to release application 6-7/GI/16274 from the Estonian Biobank.

The cases included in the German GWAS were approved by the ethics committees of the University of Bonn and the Technical University of Munich (KaBI DHM). The control groups were drawn from the following biobanks, each approved by their respective local ethics committees: the Heinz Nixdorf Recall Study (University Hospital Essen), the PROCAM-2 Study (University of Münster), as well as the PopGen Biobank and the FOCUS Study (University of Schleswig-Holstein). All participants provided written informed consent.

Participants were recruited by HerediGene and Inspire studies. HerediGene is a population study, a large-scale collaboration between Intermountain Healthcare, deCODE genetics and Amgen. Inspire is Intermountain's active registry for the collection of biological samples, clinical information, laboratory data and genetic information, from consenting patients diagnosed with any healthcare-related conditions. The Intermountain Healthcare IRB approved both studies, and all participants provided written informed consent before enrollment.

At Mass General Brigham Biobank, all participants provided written/electronic informed consent for broad biological and genetic research. The study protocol to analyze MGBB data was approved by the Mass General Brigham IRB under protocol 2018P001236. At UCLA ATLAS, all individuals provided written informed consent to participate in the study. Patient Recruitment and Sample Collection for Precision Health Activities at UCLA is an approved study by the UCLA IRB (17-001013). The TIMI trials were approved by each site's IRB or ethics committee, including protocols for genetic analyses.

Biospecimens and associated data used in the Colorado Center for Personalized Medicine (CCPM) study were obtained from the biobank at the University of Colorado Anschutz Medical Campus (CU AMC). All samples and data were collected under IRB-approved protocol (15-0461) with appropriate informed consent from participants. Research using these materials was conducted in accordance with the ethical guidelines and regulations governing human subjects research, upholding the principles of beneficence and nonmaleficence.

Recruitment to the GENCAST study in Leicester was approved as part of the Biomedical Informatics Centre for Cardiovascular Science (BRICCS) project (REC ID 09/H0406/114). For the MVP study, all participants provided informed consent under approval from the Veterans Affairs Central IRB. The study protocols for analyzing UK Biobank (UKB) data were approved under protocol 2021P002228 and conducted under UKB application 7089. At the Center for Interdisciplinary Cardiovascular Sciences, all individuals provided written informed consent to donate valve tissue and cells for research purposes. Experimental work at the Cardiovascular Life Sciences Center is approved by the BWH IRB (2011P001703).

### Study populations and phenotyping

The IAVGC comprises 30 studies. Descriptive characteristics for contributing studies are presented in Supplementary Table 1. A consistent AS phenotype was applied across all IAVGC studies (except as

otherwise described in Supplementary Methods) using a previously validated definition for AS comprised of 'International Classification of Diseases' (ICD) and 'Current Procedural Terminology' codes[7] (Supplementary Table 2). Study-level quality control thresholds are described in detail in Supplementary Methods. Most studies performed genome-wide imputation with the NHLBI Trans-Omics for Precision Medicine (TOPMed) imputation panel[42]. After quality control and imputation, participating studies performed a GWAS using either SAIGE[43] or REGENIE[44] for both autosomes stratified by genetic ancestry, as well as autosomes and the X chromosome, stratified by sex.

## Meta-analysis

GWAS summary data were uploaded to central servers at the Broad Institute and the Digital Research Alliance of Canada and consortium-level quality control, including the removal of variants with imputation quality of ≤0.3 or minor allele count of <10, was performed independently by two authors (A.M.S. and L.D.) of this study. Summary statistics in hg19 were converted to hg38 using LiftOver (v1.04.00). LD score regression intercepts were calculated for each GWAS using LD score (v.1.0.1)[45] and corrected standard errors (SEldsc) were calculated by multiplying the standard error by the square root of the LD score regression intercept in cases where the LD score regression intercept was more than 1. Fixed-effects, inverse-variance weighted meta-analysis was performed using GWAMA (v2.2.2)[46] with SEldsc to correct for inflation. GWAS meta-analysis was performed for the entire multi-ancestry population as well as for ancestry-stratified and sex-stratified populations. X chromosome analysis was performed by meta-analyzing all sex-stratified X chromosome data. Variants with an MAF of ≥0.001 and present in only one study or with an MAF of <0.001 and present in three or fewer studies were removed from the resulting meta-analysis summary files. Genome-wide significance was defined as $P < 5 \times 10^{-8}$. Independent lead variants in each GWAS were established by determining the top-most significant variant within a 500-kb region. Lead variants were additionally tested for independence by establishing that each lead variant was independent ($r^2 < 0.2$) from all other lead variants in all available 1000 Genomes (1000G) populations. All variant pairs with an $r^2$ between 0.1 and 0.2 were additionally evaluated for conditional independence using European genetic ancestry individual-level data in the MVP. Variant pairs were evaluated in association with AS independently, and, in a joint model, adjusting for age[2], sex and principal components. For variant pairs that were not conditionally independent, the top-most significant variant of the pair was considered the lead variant. Random-effects inverse-variance weighted meta-analysis was performed as a sensitivity analysis for lead variants with significant heterogeneity ($q$ value < 0.05/261 (total number of independent lead variants) = 0.0002). Liability-scale heritability was calculated using LD score (v.1.0.1). Percent variance explained was calculated from independent lead variants using the method described in ref. 47.

## Sex interaction

We tested for differences in AS effect sizes between males and females for lead variants identified in our multi-ancestry meta-GWAS using[48]:

$$Z = \frac{B_m - B_f}{\sqrt{\text{s.e.}_m^2 - \text{s.e.}_f^2 - 2 \times r \times \text{s.e.}_m \times \text{s.e.}_f}}$$

where $B_m/\text{s.e.}_m$ refers to the $\beta/\text{s.e.}$ in the AS male meta-GWAS, $B_f/\text{s.e.}_f$ refers to the $\beta/\text{s.e.}$ in the AS female meta-GWAS, and $r$ refers to the correlation between $\beta$ in the AS male and female meta-GWAS. We considered effect estimates to be significantly different by sex if the $z$ score for the difference was greater than 3.7 (corresponding to a two-tailed $P$ value of <0.05/252 (total number of independent lead variants in multi-ancestry or ancestry-stratified GWAS) = $2.0 \times 10^{-4}$).

We also evaluated whether any lead variants identified in the male or female AS meta-GWASs were independent of those discovered in

our combined meta-analysis. We considered sex-specific lead variants independent from combined GWAS lead variants if they were both greater than 500 kb from any full population lead variant and in linkage equilibrium ($r^2 < 0.2$) with all full multi-ancestry population lead variants in all 1000G populations. We then evaluated for heterogeneity between the male and female sex-stratified GWAS using a fixed-effects inverse-variance weighted meta-analysis framework for lead variants identified in sex-stratified GWAS ($n = 5$). Significant heterogeneity was considered for $P < 0.05/5 = 0.01$.

## Transcriptome-wide association analysis

Transcriptomic data were previously generated from human AV samples from 484 individuals who underwent AV replacement or heart transplant at the Institut Universitaire de Cardiologie et de Pneumologie de Québec-Université Laval (IUCPQ-UL) as part of the QUEBEC-CAVS study[9]. All participants provided informed consent and the study was approved by the ethics committee of the IUCPQ-UL. Briefly, RNA sequencing was performed on a NovaSeq 6000 instrument (Illumina), targeting >50 million paired reads per sample. Read counts were generated using GENCODE Release 41 on build GRCh38. Genotyping was performed using the Illumina Global Screening Array. All transcriptomic data were from participants who self-reported as European and clustered with the 1000G Phase 3 European ancestry data. Genotypes were imputed using the TOPMed Imputation Server with the TOPMed Imputation Reference panel (version TOPMed-r2). Variants with an MAF of <0.01 or imputation quality score of <0.3 were excluded.

We used our human AV transcriptomic data to generate a gene-expression model estimating the regulatory effects of SNPs on protein-coding gene expression using the software PredictDB (v7)[49]. Elastic-net models were trained using nested cross-validation from genotype and normalized gene expression data adjusted for age, sex, smoking status (current or not), the first 60 probabilistic estimation of expression residuals[50] factors, and the first five ancestry-based principal components. Variants were considered to have a regulatory effect on gene expression if they were located within 1 Mb of the transcription start site for any given gene of interest. A model testing the association between a given SNP and gene expression was considered significant when the average Pearson correlation between predicted and observed expression was greater than 0.1 and the estimated $P$ value was less than 0.05.

A TWAS was then performed using the S-PrediXcan extension[51] in MetaXcan (v0.7.4) with European genetic ancestry summary statistics from our autosomal meta-GWAS of AS (chosen to optimize population structure overlap between the AS GWAS and AV samples). The statistical significance threshold was set using Bonferroni correction for the number of genes tested ($P < 0.05/10,574 = 4.73 \times 10^{-6}$).

Colocalization between eQTLs in human AVs and AS risk was evaluated using COLOC (v3.2.1) for genes identified by TWAS[52] and variants from the IAVGC AS GWAS located within 1 Mb of a gene's transcription start or end sites. eQTLs were generated using QTLtools (v1.1)[53]. The two signals were considered colocalized if their posterior probability of shared signal (PP4) was >0.75. The LocusCompareR package (v1.0.0) was used to validate colocalization[54].

We also compared relative gene expression between the AV and 43 GTEx[55] tissues using previously calculated ESS[9]. Briefly, ESS were calculated by dividing the median $\log_2$(transcripts per million) value from AV tissue by the sum of the median $\log_2$(transcripts per million) values of all 43 GTEx v8 tissues. An ESS of greater than 0.1 in the AV (corresponding to AV-specific gene expression of greater than 10% total gene expression in all examined tissues) was considered the threshold for significant AV gene expression enrichment.

## eQTL colocalization

We performed eQTL colocalization for all lead variants in autosomes using AS GWAS summary statistics and *cis*-QTL data from GTEx v8 for

relevant extravalvular tissues (heart left atrial appendage, heart left ventricle, lung, liver, skeletal muscle, whole blood, cultured fibroblasts, Epstein−Barr virus-transformed lymphocytes, subcutaneous adipose, visceral omentum adipose, coronary artery, tibial artery and aorta). For each colocalization analysis, AS summary data were subset to a region within 1 Mb around each lead variant, and these were merged with variant–QTL associations from tissue-specific GTEx data. Colocalization was performed using the COLOC (v5.1.0) package in R. We considered a PP4 >0.75 as evidence of colocalization.

### Combined SNP to gene

Combined SNP to gene (cS2G) leverages seven different SNP-to-gene prioritization strategies to generate an optimal SNP–gene pair per significant independent SNP[56]. We obtained cS2G annotations for all IAVGC lead variants, and restricted SNP–gene pairs with a cS2G score ≥0.5 to maximize precision/recall.

### Causal gene prioritization

For each lead variant, we generated a list of prioritized genes based on the following methods: (1) nearest gene, (2) cS2G, (3) extravalvular eQTL colocalization, (4) protein-altering variation, (5) AV eQTL TWAS and eQTL colocalization and (6) AV gene expression or protein abundance data. We considered a lead variant to be prioritized by protein-altering variation if it was in significant LD ($r^2 > 0.8$) with a protein-coding variant. Coding variants were further annotated as damaging if they were missense and predicted by PolyPhen-2 (ref. [57]) to be probably damaging or by SIFT[58] to be deleterious or were protein truncating. We considered genes to be prioritized based on human AV transcriptomic data if they were both significant in TWAS ($P < 4.73 \times 10^{-6}$) and in eQTL colocalization (PP4 > 0.75). AV protein abundance and gene expression data were obtained from published liquid chromatography–mass-spectrometry-based proteomics and transcriptomics datasets from human AV tissue and cultured VICs[18,19]. In the dataset discussed in ref. [19], nine human AV specimens from patients with severe AS were obtained and microdissected into nondiseased, fibrotic and calcific segments of the valve. Mass spectrometry proteomics ($n = 9$) and transcriptomics ($n = 3$) were performed comparing diseased and nondiseased segments. Genes were annotated based on whether the protein product was detected in bulk proteomics, whether the gene transcript was identified in bulk transcriptomics, or whether protein abundance/gene expression was differentially apparent across disease states (defined as adjusted $P < 0.5$ and absolute $\log_2$(fold change) > 0.5). VICs were cultured from these samples (separately from the fibrosa and ventricularis) and subjected to either osteogenic or normal media (NM) conditions. Mass spectrometry proteomics was then performed to compare protein abundances among VICs in osteogenic and NM conditions. In the dataset discussed in ref. [18], human AV specimens were obtained from patients with severe AS and microdissected into nondiseased, fibrotic and calcific segments. Mass spectrometry proteomics was then performed comparing diseased (calcific or fibrotic) and nondiseased segments. Genes were annotated based on whether their protein products were detected in bulk proteomics or were differentially expressed across disease states (defined as adjusted $P < 0.5$ and absolute $\log_2$(fold change) > 0.5).

A single causal gene was determined for all lead variants using the following criteria: (1) for lead variants with significant LD ($r^2 > 0.8$) to a damaging protein-altering variant, the altered gene was prioritized as the most likely causal gene, (2) by consensus of the greatest number of indicators (including nearest gene, cS2G, extravalvular eQTL colocalization, nondamaging protein-altering variation, AV eQTL TWAS and eQTL colocalization, detection in AV proteomics, or differential gene expression in AV transcriptomics) or (3) for variants with only one indicator or with equal numbers of indicators for more than one gene, the nearest gene was prioritized.

### Gene-set enrichment

We used DEPICT (v1) to prioritize causal gene sets from our multi-ancestry AS GWAS summary statistics[59]. We defined significant gene-set enrichment for results with an enrichment $P$ value $<0.05/10,968 = 4.6 \times 10^{-6}$. We then created similarity matrices among gene sets using the Jaccard index and performed affinity propagation with the apcluster package (v1.4.11) in R, following the method described in ref. [60]. Exemplar gene sets were then plotted as nodes with the density of edges representing the similarity of genes between sets. We additionally annotated causal gene sets using Enrichr, a web-based software that prioritizes gene-set ontologies from a provider-designated list of genes[61].

### Evaluation of pleiotropy

We evaluated pleiotropic associations among all 261 independent lead variants using publicly available summary statistics from the recent PheWAS across 44.3 million genotyped variants in the MVP[20]. We queried European genetic ancestry PheWAS summary data for our *trans*-ancestry meta-analysis lead variants and European genetic ancestry lead variants, and African genetic ancestry PheWAS summary data for our African genetic ancestry lead variants. We evaluated associations across all 1,854 binary and 214 quantitative traits in the MVP PheWAS and considered any association with a $P$ value $<9.3 \times 10^{-8}$ to be statistically significant, which is a Bonferroni-corrected significance threshold accounting for 261 lead variants evaluated across 214 quantitative and 1,854 binary traits.

### Development of an AS PRS

We developed an AS PRS using summary data from autosomal AS GWAS meta-analysis performed in a sample excluding the MGBB and UKB populations, which were used to test and validate the PRSs, respectively. PRSs were developed using LDpred2 (ref. [62]) and PRS-CS[63], both of which use Bayesian approaches using GWAS summary-level data. Autosomal Hapmap3 SNPs were extracted from multi-ancestry and population-specific AS GWAS and used as the inputs of LDpred2 and PRS-CS. LD reference panels for both LDpred2 and PRS-CS were built using 1000G data of matched populations for each population-specific AS meta-GWAS (European, African, Hispanic, East Asian and South Asian). We used a European LD reference panel for our multi-ancestry AS meta-GWAS as the majority of samples in the multi-ancestry IAVGC AS meta-GWAS were European.

For LDpred2, we generated multiple PRSs using a grid of hyperparameters, including assumed heritability of 0.7, 1.0 and 1.4 times the estimated heritability, assumed proportion of causal SNPs as a sequence of 17 values from $1 \times 10^{-4}$ to 1 on a log-scale, and sparsity (true or false, representing whether some of the posterior effect size can be shrunk to zero). For PRS-CS, we used the default hyper-parameter settings indicated by the authors $- a = 1$, $b = 0.5$, $\varphi = 1 \times 10^{-6}$, $1 \times 10^{-4}$, $1 \times 10^{-2}$, 1. The resulting PRSs were tested using European data from MGBB, a nonoverlapping dataset. For each set of posterior effect sizes generated using either LDpred2 or PRS-CS, we identified the PRS in the MGBB with the best predictive value (highest phenotypic variance explained by $r^2$).

### AS PRS evaluation

The best-performing PRS in the MGBB was validated using data from the UKB, UCLA ATLAS[64], and aggregate data from six TIMI clinical trials (ENGAGE AF-TIMI 48 (ref. [65]), SOLID-TIMI 52 (ref. [66]), SAVOR-TIMI 53 (ref. [67]), PEGASUS-TIMI 54 (ref. [68]), DECLARE-TIMI 58 (ref. [69]) and FOURIER (TIMI 59) (ref. [70])), all of which are independent samples from those used for the AS GWAS. Cox proportional hazards models were used to calculate HRs in both the UKB and TIMI trials for AS against our continuous, normalized AS PRS in an analysis adjusting for age, sex, genetic ancestry principal components 1–5 and clinical risk factors including T2D, HTN, CAD, HLD, body mass index, current smoking and renal failure (eGFR < 30 ml min$^{-1}$ 1.73 m$^{-2}$). Testing using TIMI trial data

required that individual patient-level data were pooled from the six clinical trials. All analyses were compared to the performance of our previously published AS PRS generated using MVP data[21]. Results from the UKB and TIMI clinical trials were meta-analyzed using fixed-effects, inverse-variance weighting. Logistic regression was used to calculate ORs in UCLA ATLAS for AS against our continuous, normalized AS PRS with the same covariates included for the UKB and TIMI trial analyses. We also evaluated whether the AS PRS was associated with incident AV replacement in the UKB, using a composite outcome of surgical or transcatheter AV replacement codes, compared with a control population without any AS. We assessed AS risk prediction for genetic and clinical factors in the UKB using a Cox proportional hazards model including genetic risk categories (top 1%, 2%, 10% and 20% of genetic risk, compared to a referent of middle 40–60% genetic risk), adjusting for age (>65 years), male sex, ancestry-specific principal components, T2D, HTN, CAD, HLD, elevated body mass index ($\geq 30$ kg m$^{-2}$), current smoking and renal failure (eGFR < 30 ml min$^{-1}$ 1.73 m$^{-2}$).

Phenotyping for AS and clinical risk factors in TIMI trials have been previously described[21]. Phenotyping for AS in both the UKB and UCLA ATLAS also used our IAVGC definition as stated above. Individuals with prevalent AS in the UKB were excluded. To evaluate the relative contributions of the AS PRS and individual clinical risk factors, C-indices were calculated for the AS PRS and clinical risk factors either alone or in a full model including both. The C-index was compared across models using likelihood-ratio tests. We also calculated continuous NRIs comparing models with the AS PRS and clinical risk factors to a model with clinical risk factors alone in both the UKB and TIMI trials. Kaplan–Meier curves were drawn using UKB and TIMI clinical trial data, stratified by quintiles of genetic risk.

## Isolation of human VICs

Human AV samples were collected from 11 donors undergoing AV replacement surgeries for severe AS at Brigham and Women's Hospital after written informed consent was obtained (BWH IRB protocol 2011P001703). The AV samples were kept on ice in DMEM culture media (Thermo Fisher Scientific, 11-965-118) and then washed thrice in PBS. Human primary VICs were isolated from the AV leaflets using collagenase digestion. After cutting into 1–2-mm pieces, sections were digested using 1 mg ml$^{-1}$ collagenase (MilliporeSigma, C5894) in DMEM at 37 °C for 1 h with gentle mixing every 20 min. Valvular endothelial cells were washed away with DMEM and discarded. AV pieces were further digested using 1 mg ml$^{-1}$ collagenase for 3 h with gentle mixing every 20 min and isolated VICs were collected by centrifugation at 523$g$ (1,500 rpm) for 5 min and plated in 75 cm$^2$ culture flasks. Isolated VICs were cultured in growth media (GM) containing DMEM supplemented with 10% FBS, 1% penicillin–streptomycin (PS; Lonza, 17-602E), and 1 mmol l$^{-1}$ sodium pyruvate (Thermo Fisher Scientific, 11-360-070) in a CO$_2$ incubator (37 °C, 5% CO$_2$) until the cells were >90% confluent. Then, cells were detached using 0.05% trypsin–ethylenediaminetetraacetic acid (Thermo Fisher Scientific, 25200056) and plated for subculture. VIC passages 4–7 were used for all experiments.

## Gene silencing and calcification detection in human VICs

Human VICs were plated in 24-well or 48-well plates at a density of $1 \times 10^5$ cells per ml using GM. After 24 h, cells were transfected with 20 nmol l$^{-1}$ siRNA of either LTBP4 (Horizon Discovery, L-019552-00-0005), CMKLR1 (Horizon Discovery, L-005467-00-0005), CLCA2 (Horizon Discovery, L-003813-00-0005), CERS2 (Horizon Discovery, L-010282-00-0005) or CEP120 (Horizon Discovery, L-016493-02-0005) and control (Horizon Discovery, P-001810-10-05) using DharmaFECT 1 Transfection Reagent (Horizon Discovery, T-2001-03). After 3 days, GM was replaced with NM or osteogenic media (OM), and this time point was considered as day 0. Furthermore, OM was composed of DMEM supplemented with 10% FBS, 1% PS, 1 mmol l$^{-1}$ sodium pyruvate, 10 nmol l$^{-1}$ dexamethasone,

10 mmol l$^{-1}$ β-glycerophosphate (MilliporeSigma, 35675-100G) and 100 μmol l$^{-1}$ L-ascorbic acid 2 phosphate (MilliporeSigma, A8960-5G). NM was composed of DMEM with the same concentration of FBS, PS and sodium pyruvate with GM. Media was changed every 3–4 days. siRNA transfection was performed when the media was replaced. Gene silencing by siRNA transfection was confirmed by real-time quantitative PCR (RT–qPCR).

Human VICs were suspended in 0.4-ml RNAzol (MilliporeSigma, R4533) in each well of a 24-well plate, and total RNA was extracted by following the manufacturer's instructions. In total, 160 μl RNase-free water was added and mixed for 15 s. Samples were incubated at room temperature for 5 min, and centrifuged at 12,000$g$ (10,000 rpm) for 15 min at 4 °C. The upper supernatant was transferred to a new 1.5-ml tube, leaving a layer of the supernatant above the DNA/protein pellet. An equal volume of isopropanol was added to precipitate mRNA, and the samples were incubated at room temperature for 10 min and centrifuged at 12,000$g$ (10,000 rpm) for 10 min. The supernatant was removed, and RNA pellet was washed twice with 160 μl of 75% ethanol (vol/vol). Samples were then centrifuged at 4,000–8,000$g$ for 1 min at room temperature. Alcohol solution was removed with a micropipette. The RNA pellet was solubilized without drying in 20 μl of RNase-free water by pipetting up and down about 30 times. RNA concentration was quantified using NanoDrop 2000 spectrometer (Thermo Fisher Scientific, ND-2000). Next, cDNA was prepared from the RNA sample using qScript RT (KIT F/CDNA SYNTHESIS QSCRIPT; Quanta BioSciences, 95047) as per the manufacturer's protocol and reverse transcription was performed using Thermal Cycler at 22 °C for 5 min, 42 °C for 30 min and 85 °C for 5 min. Prepared cDNA diluted 1:5 using RNase-free water. PerfeCTa FastMix II ROX (Quantabio, 97065) was used for RT–qPCR with QuantStudio5 real-time PCR system (Thermo Fisher Scientific, A28140) following the manufacturer's protocol. Gene-specific primers from Life Technologies were used—human GAPDH, Hs02758991_g1; human LTBP4, Hs00943217_m1; human CMKLR1, Hs01081979_s1; human CLCA2, Hs00998923_m1; human CERS2, Hs00371958_g1; human CEP120, Hs00537880_m1. Samples were normalized by endogenous human GAPDH.

Calcium deposition was detected using 2% Alizarin red staining solution (Lifeline Cell Technology, CM-0058). Human VICs were fixed with 10% formalin for 15 min and washed with distilled water. After adding Alizarin red staining solution, cells were stained for 30 min at room temperature. Excess stain was washed thrice with distilled water. Alizarin red staining was extracted using 5% formic acid and calcium content was quantified by absorbance at 450 nm. Statistical analysis was performed using Student's $t$ tests (two-tailed; paired) for comparison between two groups using Prism 10 (GraphPad). A $P$ value of <0.05 was considered significant. Biological replicates were used for calcification assays and qPCR. Two technical replicates were used for each experimental condition.

## Histological assessment of human AV tissues

Five donors of human AV samples were used for histological analysis. AV samples embedded into Optimum Cutting Temperature compound (OCT, Sakura Finetek) were cut into 7-μm serial sections using a cryostat (Leica, CM3050S) followed by immunohistochemical staining. Cryosections were fixed for 5 min in 4% paraformaldehyde solution and incubated for 1 h in blocking solution (PBS, 10% donkey serum, 1% BSA) at room temperature. Sections were then incubated with primary antibodies—anti-LTBP4 antibody (Invitrogen, PA5-85149) and anti-CMKLR1 antibody (Abcam, ab230442), overnight at 4 °C. After washing with PBS, sections were incubated with fluorescence-conjugated secondary antibodies, specifically donkey antigoat IgG (H + L) cross-adsorbed secondary antibody (Alexa Fluor 594, 1:100 dilution; Invitrogen, A-11058) for 45 minutes at room temperature, followed by two washes with PBS. Slides were then incubated with calcium-binding near-infrared imaging fluorescence agent, Osteosense680 (1:1,000) for 30 min and then

mounted with a mounting medium containing DAPI (VECTASHIELD, H-1500). The fluorescence signal was examined with a Nikon Eclipse Confocal microscope (Nikon).

## Reporting summary

Further information on research design is available in the Nature Portfolio Reporting Summary linked to this article.

## Data availability

Summary statistics from our multi-ancestry GWAS, and all stratified analyses (by ancestry and sex) are available in the CVD Knowledge Portal (https://cvd.hugeamp.org/dinspector.html?dataset=Small2025_AorticStenosis). PRSs for AS are available in the Polygenic Score Catalog (PGS005252; https://www.pgscatalog.org/). Researchers can apply for data from contributing biobanks following their data application procedures (Supplementary Table 17). We used publicly available human eQTL data from GTEx v8 at https://gtexportal.org/home/.

## Code availability

Publicly available software was used to perform the analyses. Software packages, versions, and associated URLs are as follows: R statistical software (v4.1; https://www.r-project.org/), LDSC (v.1.0.1; https://github.com/bulik/ldsc), LiftOver (v1.04.00; https://liftover.broadinstitute.org/), GWAMA (v2.2.2; https://genomics.ut.ee/en/tools), PredictDB (v7; https://predictdb.org/), MetaXcan (v0.7.4; https://github.com/hakyimlab/MetaXcan), COLOC (v3.2.1; https://chr1swallace.github.io/coloc/), QTLtools (v1.1; https://qtltools.github.io/qtltools/), LocusCompareR (v1.0.0; https://github.com/boxiangliu/locuscomparer), DEPICT (v1; https://github.com/perslab/depict), apcluster (v1.4.11; https://github.com/UBod/apcluster) and Enrichr (https://maayanlab.cloud/Enrichr/). LDPred2 was implemented through the R package bigsnpr (v1.12.2; https://privefl.github.io/bigsnpr/), bigstatsr (v1.6.1; https://privefl.github.io/bigstatsr/) and PRS-CS (4 June 2021) (https://github.com/getian107/PRScs).

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

## Acknowledgements

A.M.S. is supported by a Physician Scientist Fellowship grant from the Doris Duke Charitable Foundation. S.T. is supported by the Heart and Stroke Foundation of Canada (G-19-0026386) and Canadian Institutes of Health Research (PJT-162344). Y.B. is supported by Canadian Institutes of Health Research (MOP-102481, MOP-137058, PJT-153396 and PJT-153396) and holds a Canada Research Chair in Genomics of Heart and Lung Diseases. G.M.P. and P.N. are supported by R01HL127564. G.T. and J.C.E. are supported by the Canadian Institutes of Health Research (PJT-191931 and PJT-180447) and the Heart and Stroke Foundation of Canada (G-24-0036449 and G-22-0031961). G.T. is supported by a salary award from the Fonds de Recherche Québec - Santé.

## Author contributions

A.M.S., J.C.E., G.M.P., P.N. and G.T. designed the study. A.M.S., T.-Y.Y., S.I., S.T. and L.D. performed the experiments. R.K., I.K., K.M. and K.I. provided data from Biobank Japan. H.M.T.V., R.D. and R.J.F.L. provided data from BioME. E.H.F.-E., L.L.S. and Q.S.W. provided data from BioVU. K.M.B. and B.P. provided data from UCLA ATLAS. K.M.C., M.E.R. and S.H.S. provided data from CATHGEN. F.L., S.L.S., J.-J.S., D.M.Z. and T.L.T. provided data from CAVS-France. S.I., S.A.S., A.R.S. and E. Aikawa performed siRNA knockout and calcification assays in VICs. H.A.S., N.R. and C.G. provided data from CCPM. J.G., A.A.R., S.R.O., E.S., C.M., O.B.P., C.E., H.U. and H.B. provided data from the DBDS Genomic Consortium. D.F.G., H.H., A.H. and K.S. provided data from deCODE and Intermountain. E. Abner and T.E. provided data from the Estonian Biobank Research Team. J.L. and A.G. provided data from FinnGen. T.T., J.S., C.M., B.A.-K., G.N., M.D., M.K., K.B., M.M.N., A.F., U.N.-G. and H.A.S. provided data from the German Aortic Stenosis Cohort. S.F. and D.A.v.H. provided data from the Genes & Health Research Team. M.R.M. and B.M.B. provided data from HUNT. S.K., K.U.K., L.N., R.D., M.D.M., P.S.B., C.P.N. and N.J.S. provided data from Leicester. T.C., O.M. and J.G.S. provided data from the Malmo Diet and Cancer Study. M.S.S., S.K. and R.B. provided data from MGBB. Y.R. provided code for developing PRS. J.L. provided data from the Umea University Biobank. S.M.D., M.G.L. and D.J.R. provided data from PMBB. C.T.R., G.E.M.M., F.K.K., N.A.M. and M.S.S. provided data from TIMI clinical trials. S.C.L. and K.M. provided data from COSMC/SIMPLER/SMC. S.T., P.M. and Y.B. provided eQTL data and relevant analyses from human AVs. K.C., P.W.F.W. and C.J.O. provided data from MVP. A.M.S. and T.-Y.Y. drafted the paper. All authors reviewed the paper and provided critical revisions.

## Competing interests

Authors affiliated with deCODE genetics/Amgen are employed by the company (G.S., D.F.G., A.F., K.S., A.H. and H.H.). M.K. is a physician proctor and a member of the medical advisory board for JOMDD, a physician proctor for Peter Duschek, a medical consultant for EVOTEC and Moderna and has received speakers' honoraria from Medtronic and Terumo. S.M.D. has research support from Renalytix and Novo Nordisk. R.J.F.L. is a consultant for Eli Lilly and Novo Nordisk. S.S. receives speaker's honoraria and is on the scientific board for Johnson & Johnson. C.J.O. is employed by Novartis Pharmaceuticals. P.N. reports research grants from Allelica, Amgen, Apple, Boston Scientific, Genentech/Roche and Novartis, personal fees from Allelica, Apple, AstraZeneca, Blackstone Life Sciences, Creative Education Concepts, CRISPR Therapeutics, Eli Lilly, Esperion Therapeutics, Foresite Capital, Foresite Labs, Genentech/Roche, GV, HeartFlow, Magnet Biomedicine, Merck, Novartis, TenSixteen Bio, and Tourmaline Bio, equity in Bolt, Candela, Mercury, MyOme, Parameter Health, Preciseli, and TenSixteen Bio, and spousal employment at Vertex Pharmaceuticals, all unrelated to the present work. G.T. has received consulting fees from Ionis Pharmaceuticals and has participated in advisory boards for Amgen, Sanofi, Novartis, HLS Therapeutics and Silence, all unrelated to the current work. The other authors declare no competing interests.

## Additional information

**Correspondence and requests for materials** should be addressed to Pradeep Natarajan or George Thanassoulis.

Pradeep Natarajan

# Reporting Summary

## Statistics

For all statistical analyses, confirm that the following items are present in the figure legend, table legend, main text, or Methods section.

| n/a | Confirmed | |
|---|---|---|
| ☐ | ☒ | The exact sample size (*n*) for each experimental group/condition, given as a discrete number and unit of measurement |
| ☐ | ☒ | A statement on whether measurements were taken from distinct samples or whether the same sample was measured repeatedly |
| ☐ | ☒ | The statistical test(s) used AND whether they are one- or two-sided *Only common tests should be described solely by name; describe more complex techniques in the Methods section.* |
| ☐ | ☒ | A description of all covariates tested |
| ☐ | ☒ | A description of any assumptions or corrections, such as tests of normality and adjustment for multiple comparisons |
| ☐ | ☒ | A full description of the statistical parameters including central tendency (e.g. means) or other basic estimates (e.g. regression coefficient) AND variation (e.g. standard deviation) or associated estimates of uncertainty (e.g. confidence intervals) |
| ☐ | ☒ | For null hypothesis testing, the test statistic (e.g. *F*, *t*, *r*) with confidence intervals, effect sizes, degrees of freedom and *P* value noted *Give P values as exact values whenever suitable.* |
| ☒ | ☐ | For Bayesian analysis, information on the choice of priors and Markov chain Monte Carlo settings |
| ☒ | ☐ | For hierarchical and complex designs, identification of the appropriate level for tests and full reporting of outcomes |
| ☐ | ☒ | Estimates of effect sizes (e.g. Cohen's *d*, Pearson's *r*), indicating how they were calculated |

*Our web collection on statistics for biologists contains articles on many of the points above.*

## Software and code

Policy information about availability of computer code

| Data collection | All genetic and/or human aortic valve data was collected after informed consent according to pre-specified operating protocols for each biobank or study, which are described in detail in the Supplemental Methods. |
|---|---|
| Data analysis | Publicly available software was used to perform the analyses. Software packages, versions, and associated URLs are as follows: R statistical software 4.1 https://www.r-project.org/. LDSC 1.0.1 https://github.com/bulik/ldsc. liftOver v1.04.00 https://liftover.broadinstitute.org/. GWAMA v2.2.2 https://genomics.ut.ee/en/tools. PredictDB v7 https://predictdb.org/. MetaXcan v0.7.4 https://github.com/hakyimlab/MetaXcan. COLOC v3.2.1 https://chr1swallace.github.io/coloc/. QTLtools v1.1 https://qtltools.github.io/qtltools/. LocusCompareR v1.0.0 https://github.com/boxiangliu/locuscomparer. DEPICT v1 https://github.com/perslab/depict. apcluster v1.4.11 https://github.com/UBod/apcluster. Enrichr https://maayanlab.cloud/Enrichr/. LDPred2 was implemented through the R package bigsnpr v1.12.2 https://privefl.github.io/bigsnpr/ and bigstatsr v1.6.1 https://privefl.github.io/bigstatsr/ . PRS-CS (June 4, 2021) https://github.com/getian107/PRScs. |

For manuscripts utilizing custom algorithms or software that are central to the research but not yet described in published literature, software must be made available to editors and reviewers. We strongly encourage code deposition in a community repository (e.g. GitHub). See the Nature Portfolio guidelines for submitting code & software for further information.

# Data

Policy information about availability of data

All manuscripts must include a data availability statement. This statement should provide the following information, where applicable:

- Accession codes, unique identifiers, or web links for publicly available datasets
- A description of any restrictions on data availability
- For clinical datasets or third party data, please ensure that the statement adheres to our policy

Summary statistics from our multi-ancestry genome-wide association study, and all stratified analyses (by ancestry and sex) are available in the CVD Knowledge Portal (https://cvd.hugeamp.org/dinspector.html?dataset=Small2025_AorticStenosis). PRSs for aortic stenosis are available in the Polygenic Score Catalog (PGS005252; https://www.pgscatalog.org/). Researchers can apply for data from contributing biobanks following their data application procedures. We used publicly available human eQTL data from GTEx v8 https://gtexportal.org/home/.

# Research involving human participants, their data, or biological material

Policy information about studies with human participants or human data. See also policy information about sex, gender (identity/presentation), and sexual orientation and race, ethnicity and racism.

| | |
|---|---|
| Reporting on sex and gender | Biological sex information was determined using genotypes. Sex is used as a covariate in all genetic association experiments and sex-stratified analyses. |
| Reporting on race, ethnicity, or other socially relevant groupings | Genetic ancestry was determined in a study-specific manner. Details on ancestry determination is available in the published other socially relevant literature for all participating cohorts, which are detailed in our supplementary materials. Genetic association studies were groupings performed in ancestry stratified populations and then meta-analyzed. |
| Population characteristics | Population characteristics, including age and sex, are described by study in Supplemental Table 1. |
| Recruitment | Recruitment is study specific but mainly occurred as part of routine health care encounters after informed consent. As such, there may be selection bias in that many studies are enriched for individuals with disease (as occurs in a hospital setting). Further, diagnosis of aortic stenosis was determined by clinical coding and not routine screening. This means that there may be cases of aortic stenosis in the control population that were not properly coded or diagnosed. However we estimate that the number of missed cases is very small compared to the large (almost 3 million) number of controls and if anything would bias genetic discovery toward the null. |
| Ethics oversight | This work represents a discovery genome-wide association study of a pre-specified outcome (aortic stenosis). No treatment effects were studied; hence, there is no need for blinding in this work. |

Note that full information on the approval of the study protocol must also be provided in the manuscript.

# Field-specific reporting

Please select the one below that is the best fit for your research. If you are not sure, read the appropriate sections before making your selection.

☒ Life sciences ☐ Behavioural & social sciences ☐ Ecological, evolutionary & environmental sciences

For a reference copy of the document with all sections, see nature.com/documents/nr-reporting-summary-flat.pdf

# Life sciences study design

All studies must disclose on these points even when the disclosure is negative.

| | |
|---|---|
| Sample size | Sample size was determined based on study specific biobank recruitment. All available samples were analyzed. |
| Data exclusions | Data exclusions were specified in the phenotype definition, which is available in the supplement. |
| Replication | We implemented several measures to ensure the reproducibility of our experimental findings. All experiments were performed using biological and/or technical replicates, as appropriate, and each experiment was repeated independently using at least 5 donors of aortic valves/valve interstitial cells to confirm consistency. Detailed protocols, standardized reagents, and calibrated equipment were used throughout the study to minimize variability. All key findings were successfully replicated across independent experiments. There were no instances where findings could not be reproduced. Minor variability observed across replicates fell within expected experimental ranges and did not affect the overall conclusions. Therefore, we confirm that all attempts at replication were successful, supporting the robustness and reliability of our results. |
| Randomization | This work represents a discovery genome-wide association study of a pre-specified outcome (aortic stenosis). No treatment effects were studied; hence, there is no need for randomization in this work. |
| Blinding | This work represents a discovery genome-wide association study of a pre-specified outcome (aortic stenosis). No treatment effects were |

| Blinding | studied; hence, there is no need for blinding in this work. |
|---|---|

# Reporting for specific materials, systems and methods

We require information from authors about some types of materials, experimental systems and methods used in many studies. Here, indicate whether each material, system or method listed is relevant to your study. If you are not sure if a list item applies to your research, read the appropriate section before selecting a response.

## Materials & experimental systems

| n/a | Involved in the study |
|---|---|
| ☒ | ☐ Antibodies |
| ☒ | ☐ Eukaryotic cell lines |
| ☒ | ☐ Palaeontology and archaeology |
| ☒ | ☐ Animals and other organisms |
| ☐ | ☒ Clinical data |
| ☒ | ☐ Dual use research of concern |
| ☒ | ☐ Plants |

## Methods

| n/a | Involved in the study |
|---|---|
| ☒ | ☐ ChIP-seq |
| ☒ | ☐ Flow cytometry |
| ☒ | ☐ MRI-based neuroimaging |

## Clinical data

Policy information about clinical studies

All manuscripts should comply with the ICMJE guidelines for publication of clinical research and a completed CONSORT checklist must be included with all submissions.

| Clinical trial registration | NA |
|---|---|
| Study protocol | NA |
| Data collection | Data collection is study specific and described in the supplement. |
| Outcomes | Outcomes collection is study specific and described in the supplement. |

## Plants

| Seed stocks | *Report on the source of all seed stocks or other plant material used. If applicable, state the seed stock centre and catalogue number. If plant specimens were collected from the field, describe the collection location, date and sampling procedures.* |
|---|---|
| Novel plant genotypes | *Describe the methods by which all novel plant genotypes were produced. This includes those generated by transgenic approaches, gene editing, chemical/radiation-based mutagenesis and hybridization. For transgenic lines, describe the transformation method, the number of independent lines analyzed and the generation upon which experiments were performed. For gene-edited lines, describe the editor used, the endogenous sequence targeted for editing, the targeting guide RNA sequence (if applicable) and how the editor was applied.* |
| Authentication | *Describe any authentication procedures for each seed stock used or novel genotype generated. Describe any experiments used to assess the effect of a mutation and, where applicable, how potential secondary effects (e.g. second site T-DNA insertions, mosiacism, off-target gene editing) were examined.* |

