## [Peer Review File · Nature Genetics]

Genomic and Transcriptomic Analyses of Aortic Stenosis Enhance Therapeutic Target Discovery and Disease Prediction

Corresponding Author: Dr George Thanassoulis

Version 0:

Decision Letter:

29th Oct 2024

Dear Dr Thanassoulis,

Your Article, "Novel Insights into the Genesis of Aortic Stenosis from Genomic and Transcriptomic Data for Therapeutic Target Discovery and Disease Prediction" has now been seen by 2 referees. You will see from their comments below that while they find your work of interest, some important points are raised. We are interested in the possibility of publishing your study in Nature Genetics, but would like to consider your response to these concerns in the form of a revised manuscript before we make a final decision on publication.

To guide the scope of the revisions, the editors discuss the referee reports in detail within the team with a view to identifying key priorities that should be addressed in revision. In this case, we think both referees have provided constructive reviews aimed at strengthening the analyses and improving the presentation, and we particularly ask that you address their technical comments as thoroughly as possible with appropriate revisions. We hope that you will find the prioritized set of referee points to be useful when revising your study. Please do not hesitate to get in touch if you would like to discuss these issues further.

We therefore invite you to revise your manuscript taking into account all reviewer and editor comments. Please highlight all changes in the manuscript text file. At this stage we will need you to upload a copy of the manuscript in MS Word .docx or similar editable format.

*2) If you have not done so already please begin to revise your manuscript so that it conforms to our Article format instructions, available

[here](http://www.nature.com/ng/authors/article_types/index.html).

*3) Include a revised version of any required Reporting Summary: <https://www.nature.com/documents/nr-reporting-summary.pdf>

Link Redacted

We hope to receive your revised manuscript within 3 to 6 months. If you cannot send it within this time, please let us know.

Sincerely,

Wei Li, PhD
Senior Editor
Nature Genetics
www.nature.com/ng

Reviewers' Comments:

Reviewer #1 (Remarks to the Author):

Small and colleagues report data on the largest GWAS on aortic stenosis. Follow-up analyses are comprehensive including silencing experiments of prioritized genes. The data are very well presented. The study is a very important contribution to the field of aortic stenosis genomics.

I have a single major comment:

- The PRS analyses are interesting and confirm prior studies. The authors now derived a new PRS that outperforms a previous one. Nonetheless, I feel that more refined analyses should be undertaken and would be needed for potential clinical implementation (or for identifying high risk patients to be enrolled in clinical trials of AS prevention/modifying drugs).

a) The threshold used to define high risk is arbitrary. Why was the validation population stratified by lowest 20%, mid 60% and highest 20% PRS? PRS studies have traditionally stratified populations into high risk and non-high risk, generally defined as top 2-10% vs. bottom 90-98% (e.g. PMID 38374346). Can the authors show the HRs using different PRS thresholds (e.g. top 20-10-5-2-1%). Comparisons should be made with the rest of the population, instead of the lowest group to avoid magnifying the effect as in Figure 4B. The authors can also consider defining a high risk threshold from their testing set (MGBB) using a published optimal cutpoint metric (e.g. youden index) and testing it in the validation sets (UKB and TIMI).

b) The authors' data suggest that the performance of the clinical predictors is excellent in the UKB (e.g. Supp Figure 13A). Whether PRS adds anything to the clinical prediction remains unclear. The authors do mention on lines 643-644 that adding PRS to clinical predictors improves the C-index and provide a P-value <0.0001. It is not clear what the p-value refers to, perhaps a delong test? In addition to comparing discrimination (C-index; delong test), the authors should provide more in-depth analyses to compare clinical prediction vs. clinical+genetic prediction, such a log likelihood ratio test, changes in model calibration, changes in net clinical benefit, reclassification index, etc.

c) Provide KM curves for the tested PRS for all included testing/validation test sets.

d) If possible, provide analyses restricting the outcome to AS requiring an intervention (e.g. removing ICD codes I35.0, 424.1 from the definition).

Minor comments:

- Figure S1: Proteome wide MR is listed twice.

- The rationale for the lookup of rs10455872 in diverse ancestries in lines 558-562 and Table S6 is not clear. Isn't the lack of significance in Hispanics and South Asians simply due to limited sample size? The authors should briefly explain the

rationale in the results section or remove altogether. I would suggest the latter.

- Figure 3: Clarify in legend that XX loci mapped to genes with less than 3 supportive predictors are not shown.
- Table S14: Clarify in the legend what the HR refers to (HR per SD increase in PRS?)
- The authors tested multiple PRS in the validation set (MGB) and selected the best for further analyses. Consider providing C-index of all PRS in the validation set in a supplementary table.
- It is not clear how those 5 genes were prioritized: CERS2, CEP120, LTBP4, CMKLR1, CLCA2. This sentence is not clear "which had strong evidence for biologic relevance in AS, defined as genes with a genome-wide significant coding variant predicted to be pathogenic (Supplemental Table 11).
- Lines 681-682 "with a 2-fold increase in association metrics with AS compared to any previously published" This should be rephrased. See major comment 1 above.

Reviewer #2 (Remarks to the Author):

Summary

The authors performed a large multi-ancestry GWAS for aortic stenosis (AS) comprising over 80,000 cases and 2.8M controls with reporting of 261 loci and 223 as novel. Additional analyses were done for sex-specific and ancestry specific loci. To identify further loci and identify candidate genes transcriptomic data was leveraged from aortic valve samples, alongside several bioinformatic analyses for gene identification. Polygenic risk scores with the new loci demonstrated increased metrics in comparison a prior PRS by this group and functional experiments provided supported for additional studies on two of the 5 candidate genes studied.

This is a comprehensive study to identify loci for AS and candidate genes, with a lot of work and nice to see some experimental assays to follow up select candidate genes. The paper is well written and easy to follow. I have a few comments for the authors to consider.

Comments for the authors

1. As part of the introduction can the phenotype be expanded and the term of calcific aortic stenosis be introduced, and further references on what is known about the genetic basis of AS before this study?
2. Can you comment on phenotype of AS in your collaboration if it is the same criteria that is used for all genetic studies of this phenotype?
3. Methods – can you expand on the ESS method?
4. You report many new loci, it would be interesting to include a table of results of lookups of the lead variants for AS with other phenotypes, in the discussion there is reporting of some candidate genes, and there is overlap, knowing this across other traits will be of interest to the reader and a brief discussion.
5. Can you expand on the Schlotter dataset and VICs?
6. Can you include the heritability of AS and from your results the percent variance explained, latterly when comparing the PRSs can a comment also be included for the PRS developed from the MVP GWAS?
7. Typo HLD should be HDL – page 20, line 434
8. Can I check 54M variants were included in the GWAS?
9. I note reporting of loci – in the abstract it may be helpful to indicate number of 241 variants and what is novel comes from a range of different analyses, this could be clearer.
10. I also wondered about claiming of novel loci and the lack of stringency of the r^2 threshold you are using. How many are novel if you reduce the r^2 threshold to 0.1 or 0.2 that is usually the criteria used alongside distance. Did you account for loci recently reported by Theriault et al, Nature Comms in 2024 in their paper Integrative genomic analyses identify candidate causal genes for calcific aortic valve stenosis involving tissue-specific regulation. <https://www.nature.com/articles/s41467-024-46639-4>
11. Enrichr – not mentioned in methods – can you add please?
12. PRS tested in 3 different study settings – in the text there is some repetition of methods in results section, can this be revised. State what was the best performing PRS in the results, this is currently missing.
13. Later in the discussion I think it is worth commenting on the differences in the C index for clinical risk factors between UKB and TIMI (0.85 and 0.77), noting added value with the PRS.
14. Gene silencing – curiosity on the selection of genes was done on missense variation and gene expression in relevant tissue, why this selection criteria considering numerous lines of evidence from bioinformatic tools across the paper.
15. Can I check the MAF cut off for the GWAS, as genes with rare missense variants taken into functional testing, what was the MAF as I had thought 1% was used for the GWAS.

Version 1:

Decision Letter:

Our ref: NG-A66552R

13th Mar 2025

Dear Dr. Thanassoulis,

Thank you for submitting your revised manuscript "Novel Insights into the Genesis of Aortic Stenosis from Genomic and Transcriptomic Data for Therapeutic Target Discovery and Disease Prediction" (NG-A66552R). It has now been seen by the original referees and their comments are below. The reviewers find that the paper has improved in revision, and therefore we'll be happy in principle to publish it in Nature Genetics, pending minor revisions to satisfy the referees' final requests and to comply with our editorial and formatting guidelines.

Sincerely,
Wei

Wei Li, PhD
Senior Editor
Nature Genetics
www.nature.com/ng

Reviewer #1 (Remarks to the Author):

The authors appropriately addressed my comments/suggestions.
The manuscript is of excellent quality, and significantly adds to the available literature on the genomics and mechanisms of aortic valve stenosis.

Reviewer #2 (Remarks to the Author):

Thank you to the authors for their revised paper which provides a good response to all the queries I raised. I appreciate the time and effort, and the additions provide a really comprehensive story of your work and the main observations.

I have only one minor comment. Check for consistency in labelling in the paper transancestry and multi ancestry? Trans is mentioned in abstract.

Reviewer #1 (Remarks to the Author):

Small and colleagues report data on the largest GWAS on aortic stenosis. Follow-up analyses are comprehensive including silencing experiments of prioritized genes. The data are very well presented. The study is a very important contribution to the field of aortic stenosis genomics.

We thank the reviewers for their careful consideration of our manuscript.

I have a single major comment:

- The PRS analyses are interesting and confirm prior studies. The authors now derived a new PRS that outperforms a previous one. Nonetheless, I feel that more refined analyses should be undertaken and would be needed for potential clinical implementation (or for identifying high risk patients to be enrolled in clinical trials of AS prevention/modifying drugs).

a) The threshold used to define high risk is arbitrary. Why was the validation population stratified by lowest 20%, mid 60% and highest 20% PRS? PRS studies have traditionally stratified populations into high risk and non-high risk, generally defined as top 2-10% vs. bottom 90-98% (e.g. PMID 38374346). Can the authors show the HRs using different PRS thresholds (e.g. top 20-10-5-2-1%). Comparisons should be made with the rest of the population, instead of the lowest group to avoid magnifying the effect as in Figure 4B. The authors can also consider defining a high risk threshold from their testing set (MGBB) using a published optimal cutpoint metric (e.g. youden index) and testing it in the validation sets (UKB and TIMI).

Thresholds for continuous biomarkers are frequently used in clinical medicine, and we agree that there is no consensus yet on the appropriate threshold for an aortic stenosis polygenic risk score. For other polygenic risk scores, various thresholds have been used in the literature including the top 20th percentile¹⁻³. To address the request for effect estimates for additional thresholds, we now provide results using the top 20th, 10th, 5th, 2nd, and 1st percentiles of the genetic risk score compared to a referent of 40-60% genetic risk. These analyses are depicted in a revised **Figure 4B** as well as accompanying revised text in the methods section. Overall, these new data show, as expected, an additionally rising trajectory of risk with higher percentiles of the AS PRS. Further, all tested percentiles of the AS PRS had a higher risk estimate for AS than any clinical risk factor except for age.

Methods

489 We assessed AS risk prediction for genetic and clinical factors in the UKB using a Cox proportional hazards model including genetic risk categories (top 1%, 2%, 10%, and 20% of genetic risk, compared to a referent of middle 40-60% genetic risk), adjusting for age (> 65 years), male sex, ancestry specific principal components, T2D, HTN, CAD, hyperlipidemia, elevated BMI ($\geq 30 \text{ kg/m}^2$), current smoking, and renal failure (eGFR < 30 mL/min/1.73m²).

720 In a multivariable model including clinical risk factors, high genetic risk (the top 20th percentile or above of the PRS compared to individuals with a PRS in the 40-60th percentile) had an estimated HR greater than all other risk factors, except age > 65 years (Figure 4B). Further, there was a rising trajectory of risk with higher percentiles of the PRS.

966 Figure 4: Performance of aortic stenosis polygenic risk score in European genetic ancestry individuals from the UK Biobank, TIMI trials, and UCLA ATLAS

B) Forest plot comparing the hazard ratio (HR) for categories of genetic risk and individual risk factors in the UKB

b) The authors' data suggest that the performance of the clinical predictors is excellent in the UKB (e.g. Supp Figure 13A). Whether PRS adds anything to the clinical prediction remains unclear. The authors do mention on lines 643-644 that adding PRS to clinical predictors improves the C-index and provide a P-value <0.0001. It is not clear what the p-value refers to, perhaps a delong test? In addition to comparing discrimination (C-index; delong test), the authors should provide more in-depth analyses to compare clinical prediction vs. clinical+genetic prediction, such a log likelihood ratio test, changes in model calibration, changes in net clinical benefit, reclassification index, etc.

We compared the C-index between clinical risk prediction versus clinical + genomic risk prediction using a likelihood ratio test and have now added language to clarify this in the revised manuscript.

To provide an additional comparison between clinical risk prediction and clinical + genomic risk prediction, we calculated a continuous net reclassification index in the UKB and in TIMI clinical trials. Both the comparison of calculated C-index values and the continuous net reclassification indices clarify that the addition of the AS PRS to clinical risk factors in risk estimation of AS provides modest additional benefit. We revised the manuscript with these additional statements.

501 We also calculated continuous net reclassification indices (NRI) comparing models with the AS PRS and clinical risk factors to a model with clinical risk factors alone in both the UKB and TIMI trials.

724 The C-index for clinical risk factors (age, sex, CAD, T2D, hyperlipidemia, renal failure, current smoking, obesity) in the UKB was 0.85 [0.85-0.86], which improved to 0.87 [0.86-0.88] with the addition of the AS PRS (likelihood ratio test p-value for difference < 0.0001, Supplemental Figure 15A). The C-index for clinical risk factors alone in TIMI trials was 0.71 [0.68-0.75], which improved to 0.76 [0.72-0.79] with the addition of the AS PRS (likelihood ratio test p-value for difference < 0.0001, Supplemental Figure 15B).

731 Continuous net reclassification indices calculated between models with risk factors alone versus models with risk factors and the AS PRS were 29% (95% CI 20%-37%) in the UKB and 23% (95% CI 15%-30%) in TIMI clinical trials.

855 The addition of genetic risk to combined clinical risk factors modestly increased overall risk discrimination with a net reclassification index of 29% in the UKB and 23% in TIMI clinical trials.

c) Provide KM curves for the tested PRS for all included testing/validation test sets.

We added a supplemental figure of Kaplan Meier curves characterizing risk over time in the UKB and TIMI clinical trials, splitting genetic risk into quintiles. We were unable to generate Kaplan Meier curves for UCLA data as we did not have the ability to generate accurate

longitudinal data, requiring that we present risk using odds ratios. See the following revised text and supplemental figures.

503 Kaplan-Meier curves were drawn using UKB and TIMI clinical trial data, stratified by quintiles of genetic risk.

Supplemental Figure 14: Kaplan Meier curves depicting cumulative diagnosis of aortic stenosis by quintiles of genetic risk in the UKB and TIMI clinical trials.

A) UKB:

B) TIMI Clinical Trials

Legend: Kaplan Meier curves depicting cumulative diagnoses of aortic stenosis (Y-axis, in percent of each quintile) over time (X-axis) in the UKB (A) and TIMI clinical trials (B). Q1-Q5 represent quintile 1 (lowest 20% genetic risk) to quintile 5 (highest 20% genetic risk).

d) If possible, provide analyses restricting the outcome to AS requiring an intervention (e.g. removing ICD codes I35.0, 424.1 from the definition).

As suggested, we additionally evaluated whether our AS PRS was associated with incident aortic valve replacement in the UKB by restricting our case definition to ICD codes for valve replacement (comparing to a control population without any AS). The AS PRS had a slightly higher effect estimate in predicting aortic valve replacement relative to our primary AS definition. We have now included the revised text below.

486 We additionally evaluated whether the AS PRS was associated with incident aortic valve replacement in the UKB using a composite outcome of surgical or transcatheter aortic valve replacement codes compared to a control population without any AS.

718 In a sensitivity analysis evaluating the association between the AS PRS and risk of aortic valve replacement in the UKB (1,300 incident cases), the AS PRS was strongly associated with risk of AVR (HR 2.00 [1.89-2.12]).

Minor comments:

- Figure S1: Proteome wide MR is listed twice.

We thank the reviewer for careful review and identifying this error. Our final manuscript did not include proteome-wide Mendelian randomization and we have revised **Supplemental Figure 1** to address this. See the revised figure below.

Supplemental Figure 1: Schematic overview of study design

- The rationale for the lookup of rs10455872 in diverse ancestries in lines 558-562 and Table S6 is not clear. Isn't the lack of significance in Hispanics and South Asians simply due to limited sample size? The authors should briefly explain the rationale in the results section or remove altogether. I would suggest the latter.

We agree with the reviewers that this analysis is underpowered. Therefore, we have removed it from the manuscript.

- Figure 3: Clarify in legend that XX loci mapped to genes with less than 3 supportive predictors are not shown.

We revised the legend of Figure 3 with this revised text:

917 Legend: Schematic depicting variants with a gene prioritized by at least three methods. Green represents multi-ancestry genome-wide association study, purple represents European genetic ancestry. 154 multi-ancestry unique lead variants, 8 European genetic unique ancestry lead variants, 5 male sex unique lead variants, 1 African genetic ancestry unique lead variant, and 3 X chromosome unique lead variants with less than three supportive methods are not shown.

- Table S14: Clarify in the legend what the HR refers to (HR per SD increase in PRS?)

We revised the legend for **Supplemental Table 15 (previously Supplemental Table 14)** to clarify the hazard ratio units:

Footnote: Results for minimally adjusted (age, sex, principal components) and fully adjusted (age, sex principal components, clinical factors) prediction of AS using our novel AS PRS and previously published PRS (JAMA Cardiology 2024). Hazard ratios (HR) represent risk per 1 standard deviation of the scaled PRS.

- The authors tested multiple PRS in the validation set (MGB) and selected the best for further analyses. Consider providing C-index of all PRS in the validation set in a supplementary table.

We added a supplementary table (**Supplementary Table 14**) summarizing the Nagelkerke r^2 value for logistic regression evaluating each PRS against the binary outcome of AS for all tested PRS in MGB Biobank. We additionally commented in the results on which score had the best performance:

705 The best performing score was derived using LDpred2 with a proportion of causal SNPs of 0.032, heritability of 0.7, and with sparsity enabled. This score had a Nagelkerke r^2 for a model evaluating the PRS in prediction of AS in the MGBB of 0.05 (Supplemental Table 14).

- It is not clear how those 5 genes were prioritized: *CERS2*, *CEP120*, *LTBP4*, *CMKLR1*, *CLCA2*. This sentence is not clear "which had strong evidence for biologic relevance in AS, defined as genes with a genome-wide significant coding variant predicted to be pathogenic (Supplemental Table 11).

We appreciate the opportunity to provide additional clarity on how genes were selected for validation. We selected genes which had 1) evidence for a pathogenic role in aortic stenosis from our GWAS summary data, which we defined as genes harboring a genome-wide significant coding variant predicted to be damaging (by SIFT or PolyPhen-2), and/or genes which were the top prioritized gene in our causal gene prioritization pipeline, 2) genes which were observed to be expressed in the aortic valve using proteomics and/or transcriptomics datasets from prior human aortic valve studies, and 3) genes which were considered biologically relevant based on review by content experts and the published literature.

While we ultimately considered all data to determine whether genes were most likely to be pathogenic, we placed particular emphasis on genome-wide significant predicted damaging coding variants, as predicted damaging coding variants are expected to have a direct impact on the implicated gene product. We also required that genes have some evidence of valve-specific expression (using data from our human aortic valve transcriptome wide association study or prior proteomics/transcriptomics data). Finally, we benefitted from a literature review by content experts, including Drs. Shinsuke Ito and Elena Aikawa. We have revised the language in the results section to better clarify our selection process.

739 We performed gene silencing experiments in human VICs from 11 donors for five genes (*CERS2*, *CEP120*, *LTBP4*, *CMKLR1*, *CLCA2*) which were prioritized using the following three criteria: 1) evidence for a direct pathogenic role in aortic stenosis based on the genes harboring a genome-wide significant coding variant predicted to be damaging (Supplemental Table 11) and/or were the top prioritized gene in our causal gene prioritization pipeline (Supplemental Table 3), 2) evidence for aortic valve specific expression using data from prior human aortic valve proteomics and/or transcriptomics datasets, and 3) considered biologically relevant after review by content experts (S.I., E.A.) and the published literature.

- Lines 681-682 "with a 2-fold increase in association metrics with AS compared to any previously published" This should be rephrased. See major comment 1 above.

We revised this sentence to clarify that the observed 2-fold increase in association specifically refers to models testing the association of the PRS per standard deviation change (see **Figure 4A**).

768 Finally, we generated a novel AS PRS, with an approximately 2-fold increase in association with AS per standard deviation of the PRS, which is greater than previously published scores.

Reviewer #2 (Remarks to the Author):

Summary

The authors performed a large multi-ancestry GWAS for aortic stenosis (AS) comprising over 80,000 cases and 2.8M controls with reporting of 261 loci and 223 as novel. Additional analyses were done for sex-specific and ancestry specific loci. To identify further loci and identify candidate genes transcriptomic data was leveraged from aortic valve samples, alongside several bioinformatic analyses for gene identification. Polygenic risk scores with the new loci demonstrated increased metrics in comparison a prior PRS by this group and functional experiments provided supported for additional studies on two of the 5 candidate genes studied.

This is a comprehensive study to identify loci for AS and candidate genes, with a lot of work and nice to see some experimental assays to follow up select candidate genes. The paper is well written and easy to follow. I have a few comments for the authors to consider.

We appreciate the comments and thorough review of the manuscript.

Comments for the authors

1. As part of the introduction can the phenotype be expanded and the term of calcific aortic stenosis be introduced, and further references on what is known about the genetic basis of AS before this study?

We sincerely appreciate the attention to the phenotype. We revised the introduction to comment on the difference between calcific aortic stenosis and bicuspid/unicuspid aortic stenosis. This is the revised text in the introduction:

228 The majority of AS diagnoses are due to calcific aortic valve disease (CAVD), a fibrocalcific pathology of the aortic valve resulting in calcific AS. In contrast to AS resulting from congenital aortic valve abnormalities like bicuspid or unicuspid aortic valves, calcific AS typically occurs in older age among individuals with a trileaflet valve. Family-based studies of calcific AS estimate the heritability to be as high as 49%⁴ and several community-based studies demonstrate familial aggregation of calcific AS^{5,6}. Multiple prior genome-wide association studies (GWAS) confirm that calcific AS is a polygenic trait^{7,8}.

2. Can you comment on phenotype of AS in your collaboration if it is the same criteria that is used for all genetic studies of this phenotype?

We asked all participating studies to use the same claims-based AS phenotype previously validated in the Million Veteran Program⁷. We provide the *International Classification of Diseases* (ICD) and *Current Procedural Terminology* (CPT) codes for our AS phenotype in

Supplemental Table 2. As described in the methods section, almost all studies used our IAVGC phenotype definition (see text below). Phenotyping is described by study in the **Supplemental Methods**. The only studies that provided an alternate phenotype definition for AS were the German Aortic Stenosis Cohort (diagnosis of AS was based on echocardiography defined as a peak velocity > 2.5 m/s) and Leicester (diagnosis of AS was based on echocardiography defined as a peak velocity > 2.5 m/s or 2.0 m/s among individuals with severe left ventricular systolic dysfunction).

254 A consistent AS phenotype was applied across all IAVGC studies (except as otherwise described in the Supplemental Methods) using a previously validated definition for AS comprised of *International Classification of Diseases (ICD)* and *Current Procedural Terminology (CPT)* codes (Supplemental Table 2).

3. Methods – can you expand on the ESS method?

As requested, we revised the text to expand on the ESS method, published by Theriault et al.

360 We also compared relative gene expression between the aortic valve and 43 GTEx tissues using previously calculated expression specificity scores (ESS). Briefly, ESS were calculated by dividing the median \log_2 (transcripts per million) value from aortic valve tissue by the sum of the median \log_2 (transcripts per million) values of all 43 GTEx v8 tissues. An ESS of greater than 0.1 in the aortic valve (corresponding to aortic valve-specific gene expression of greater than 10% total gene expression in all examined tissues) was considered a threshold for significant aortic valve gene expression enrichment.

4. You report many new loci, it would be interesting to include a table of results of lookups of the lead variants for AS with other phenotypes, in the discussion there is reporting of some candidate genes, and there is overlap, knowing this across other traits will be of interest to the reader and a brief discussion.

We agree that it would be useful to characterize pleiotropic associations for our AS lead variants to better understand variant function. To comprehensively evaluate pleiotropic associations, we performed a look up for all 261 lead variants in our GWAS using publicly available phenome-wide association study (PheWAS) data for 44.3 million genotyped variants in the Million Veteran Program. We identify 220 variants with significant pleiotropy, most to lipid traits. Please see the revised text below.

439 We evaluated pleiotropic associations among all 261 independent lead variants using publicly available summary statistics from the recent Phenome Wide Association Study (PheWAS) across 44.3 million genotyped variants in the Million Veteran Program (MVP)⁹. We queried European genetic ancestry PheWAS summary data for our trans-ancestry meta-analysis lead variants and European genetic ancestry lead variants, and African genetic ancestry PheWAS summary data for our African genetic

ancestry lead variants. We evaluated associations across all 1,854 binary and 214 quantitative traits in the MVP PheWAS and considered statistically significant any association with a P-value $< 9.3 \times 10^{-8}$, which is a Bonferroni corrected significance threshold accounting for 261 lead variants evaluated across 214 quantitative and 1,854 binary traits.

684 Pleiotropic associations in aortic stenosis

We evaluated all 261 independent lead variants for pleiotropic associations across 2,068 traits including 1,854 binary and 214 quantitative traits using summary PheWAS data from the Million Veteran Program⁹. 220 lead variants (84%) had at least one pleiotropic association (Supplemental Table 14) among which 208 were trans-ancestry lead variants, 8 were European genetic ancestry lead variants, and 3 were male sex stratified lead variants. The strongest associations between continuous traits and risk of AS were among variants modifying lipid relevant measures including increased LDL cholesterol (rs6511720 [prioritized to *LDLR*], rs11591155 [*PCSK9*], rs602633 [*PSRC1*]), lower HDL cholesterol (rs116843064 [*ANGPTL4*], rs115849089 [*LPL*]), and increased triglycerides (rs115849089 [*LPL*], rs780093 [*GCKR*]). Interestingly, the second strongest continuous trait association was between lead variants associated with increased risk for AS and increased height (rs2694594 [*GLIS1*], rs11161617 [*DDAH1*]). Pleiotropic associations for binary traits are presented in Supplemental Figure 13. The top-most significant pleiotropic associations were between lead variants increasing risk for AS and risk of disorders of lipid metabolism, occurring in lead variants prioritized for *PSRC1*, *PCSK9*, *LPA*, *TRIB1*, *ZPR1*, and *LDLR*. Other notable pleiotropic associations for binary traits include the AS risk variant rs56094641 (*FTO*) and increased risk for both obesity and diabetes, and the AS risk variant rs780093 (*GCKR*) and increased risk of gout and cirrhosis.

842 The majority of independent genome-wide significant lead variants in AS had pleiotropic associations with other traits. Most notably, there were many strong associations between AS lead variants and lipid-relevant traits, including LDL cholesterol, HDL cholesterol, and triglycerides. This result is concordant with robust translational data suggesting that apolipoprotein B-containing lipoproteins are causal for AS^{7,10,11}. Several clinical trials evaluating statin therapy in mild to moderate AS failed to demonstrate efficacy in lipid-lowering as a strategy to mitigate adverse events¹²⁻¹⁴. However, these human genetic results highlight the potential role of much earlier lipid-lowering for AS risk optimization.

Supplemental Figure 13: Manhattan plot of phenome-wide association study results for aortic stenosis GWAS lead variants in the Million Veteran Program.

Legend: Manhattan plot with phenotype associations colored by trait type (see legend). Y-axis corresponds to the $-\log_{10}(p)$ of the genotype-phenotype association. X-axis corresponds to the chromosomal position of the genotype. The strongest associations are labeled by each lead variant's prioritized gene and the phenotype description.

5. Can you expand on three Schlotter dataset and VICs?

As requested, we revised the text to expand on the proteomics and transcriptomics data presented using published material from Schlotter *et al.* and Blaser *et al.*:

396 In the Schlotter *et al.* dataset, nine human aortic valve specimens from patients with severe AS were obtained and micro-dissected into non-diseased, fibrotic, and calcific segments of the valve. Mass spectrometry proteomics (n=9) and transcriptomics (n=3) were performed comparing diseased and non-diseased segments. Genes were annotated based on whether the protein product was detected in bulk proteomics, the gene transcript identified in bulk transcriptomics, or whether protein abundance/gene expression were differentially apparent between disease states (defined as adjusted p-value < 0.5 and absolute log[base2] fold change > 0.5). VICs were cultured from these samples (separately from the fibrosa and ventricularis) and subjected to either osteogenic or normal media conditions. Mass spectrometry proteomics was then performed comparing protein abundances between VICs in osteogenic and normal media conditions. In the Blaser *et al.* dataset, 18 human aortic valve specimens were obtained from patients with severe AS and micro-dissected into non-diseased, fibrotic, and calcific segments. Mass spectrometry proteomics was then performed comparing diseased (calcific or fibrotic) and non-diseased segments. Genes were annotated based on whether their protein product was detected in bulk proteomics or were differentially present between disease states (defined as adjusted p-value < 0.5 and absolute log[base2] fold change > 0.5).

6. Can you include the heritability of AS and from your results the percent variance explained, latterly when comparing the PRSs can a comment also be included for the PRS developed from the MVP GWAS?

We calculated liability-scale heritability using LDSC for our trans-ancestry meta-analysis, which we now include in the manuscript. We additionally calculated the percent variance explained using a published method by Purcell ¹⁵. We also revised the PRS results section to provide a clearer comparison of the hazard ratios between the novel AS PRS and the prior MVP AS PRS. Liability-scale heritability is notably lower than other binary cardiovascular traits (e.g. the reported liability-scale heritability for coronary artery disease and mitral valve prolapse are higher than 0.2). However, aortic stenosis typically occurs later in life and is less likely to be explained by genetics compared with mitral valve prolapse or coronary artery disease, which both manifest earlier. While not reported, we also calculated liability-scale heritability using SumHer, which resulted in a similar value (0.11). We chose to report the more conservative value with LDSC. See the revised text below.

293 Liability-scale heritability was calculated using LD Score v.1.0.1. Percent variance explained was calculated from independent lead variants using the method described by Purcell et al.¹⁵.

605 The liability-scale heritability of AS in our trans-ancestry GWAS was 0.087 and the percent variance explained from lead variants was 5.5%.

7. Typo HLD should be HDL – page 20, line 434

To clarify, we used HLD as an abbreviation for hyperlipidemia. To avoid confusion, we removed this abbreviation from the manuscript and replaced all instances with ‘hyperlipidemia’.

8. Can I check 54M variants were included in the GWAS?

Correct, while some participating studies may have had greater than, or fewer than 54 million variants in their respective GWAS, there were 54,133,673 variants in our trans-ancestry meta-analysis present in more than 1 study.

9. I note reporting of loci – in the abstract it may be helpful to indicate number of 241 variants and what is novel comes from a range of different analyses, this could be clearer.

We thank the reviewer for the comment and have revised the abstract so that the text more clearly defines the number of variants identified from our various analyses. See the revised abstract below.

198 Aortic stenosis (AS) is the most common valvular heart disease and has no pharmacological therapies. To understand the mechanisms responsible for AS, we

performed a trans-ancestry genome-wide association meta-analysis of 86,864 AS cases among 2,853,408 individuals, discovering 241 autosomal independent risk loci and three X chromosome risk loci. We additionally performed sex- and ancestry-stratified genome-wide association studies, identifying an additional five sex-specific risk loci, 11 risk loci in European ancestry individuals, and one risk locus in African ancestry individuals, bringing the total number of independent risk loci to 261, among which 223 (85%) were novel. We performed a transcriptome-wide association study using expression quantitative trait loci from human aortic valves, discovering 54 novel genes for which genetically predicted expression influences the risk of AS. Causal gene prioritization and pathway analysis of genome-wide and transcriptome-wide significant findings clarified important roles for actin and cytoskeletal pathways in modifying the risk of AS. We generated a polygenic risk score which had better risk discrimination for AS compared to clinical risk factors except age. Finally, we performed gene silencing experiments for targets of biologic relevance informed by our genome-wide association study. Silencing of *CMKLR1* and *LTBP4* in human valvular interstitial cells significantly decreased mineralization, implicating a role for polyunsaturated fatty acids and transforming growth factor beta signaling in AS pathobiology.

10. I also wondered about claiming of novel loci and the lack of stringency of the r^2 threshold you are using. How many are novel if you reduce the r^2 threshold to 0.1 or 0.2 that is usually the criteria used alongside distance. Did you account for loci recently reported by Theriault et al, Nature Comms in 2024 in their paper Integrative genomic analyses identify candidate causal genes for calcific aortic valve stenosis involving tissue-specific regulation. <https://www.nature.com/articles/s41467-024-46639-4>

We evaluated all five lead variant pairs with an r^2 between 0.1 and 0.2 for conditional independence using European genetic ancestry individual-level data in the Million Veteran Program. We found that three of the five variant pairs were conditionally independent, and therefore for these three variant pairs we kept both variants in our final count of independent lead variants. For the two variant pairs, which were not conditionally independent, we kept the stronger signal. See our **Supplemental Table 8** for details on our conditional analysis.

We did account for loci reported by Theriault and colleagues in our count of prior AS GWAS loci, which were provided to us prior to publication. We have since cited their manuscript (which was not available at the time of initial submission).

11. Enrichr – not mentioned in methods – can you add please?

We added the following text describing Enrichr:

433 We additionally annotated causal gene sets using Enrichr, a web-based software that prioritizes gene-set ontologies from a provider designated list of genes¹⁶ (<https://maayanlab.cloud/Enrichr/>).

12. PRS tested in 3 different study settings – in the text there is some repetition of methods in results section, can this be revised. State what was the best performing PRS in the results, this is currently missing.

We removed the repetitive text in the results section as suggested and added additional text describing the best-performing PRS in the results. We also included a table of the performance for all tested PRS as a reference. See the added text below as well as the new **Supplemental Table 14**.

705 The best performing score was derived using LDpred2 with a proportion of causal SNPs of 0.032, heritability of 0.7, and with sparsity enabled. This score had a Nagelkerke r^2 for a model evaluating the PRS in prediction of AS in the MGBB of 0.05 (Supplemental Table 14).

13. Later in the discussion I think it is worth commenting on the differences in the C index for clinical risk factors between UKB and TIMI (0.85 and 0.77), noting added value with the PRS.

We appreciate the comment and added the following text to the revised manuscript.

855 The addition of genetic risk to combined clinical risk factors modestly increased overall risk discrimination with a net reclassification index of 29% in the UKB and 23% in TIMI clinical trials. We note that the C-index for clinical risk factors in predicting AS was higher in the UKB compared with TIMI clinical trials, which we attribute in part to the UKB population being generally younger and healthier than individuals participating in TIMI clinical trials.

14. Gene silencing – curiosity on the selection of genes was done on missense variation and gene expression in relevant tissue, why this selection criteria considering numerous lines of evidence from bioinformatic tools across the paper.

We appreciate the opportunity to provide additional clarity on how genes were selected for validation. We selected genes which had 1) evidence for a pathogenic role in aortic stenosis from our GWAS summary data, which we defined as genes harboring a genome-wide significant coding variant predicted to be damaging (by SIFT or PolyPhen-2), and/or genes which were the top prioritized gene in our causal gene prioritization pipeline, 2) genes which were observed to be expressed in the aortic valve using proteomics and/or transcriptomics datasets from prior human aortic valve studies, and 3) genes which were considered biologically relevant based on review by content experts and the published literature.

While we ultimately considered all data to determine whether genes were most likely to be pathogenic, we placed particular emphasis on genome-wide significant predicted damaging coding variants, as predicted damaging coding variants are expected to have a direct impact on the implicated gene product. We also required that genes have some evidence of valve-specific

expression (using data from our human aortic valve transcriptome wide association study or prior proteomics/transcriptomics data). Finally, we benefitted from a literature review by content experts, including Drs. Shinsuke Ito and Elena Aikawa. We have revised the language in the results section to better clarify our selection process.

739 We performed gene silencing experiments in human VICs from 11 donors for five genes (*CERS2*, *CEP120*, *LTBP4*, *CMKLR1*, *CLCA2*) which were prioritized using the following three criteria: 1) evidence for a direct pathogenic role in aortic stenosis based on the genes harboring a genome-wide significant coding variant predicted to be damaging (Supplemental Table 11) and/or were the top prioritized gene in our causal gene prioritization pipeline (Supplemental Table 3), 2) evidence for aortic valve specific expression using data from prior human aortic valve proteomics and/or transcriptomics datasets, and 3) considered biologically relevant after review by content experts (S.I., E.A.) and the published literature.

15. Can I check the MAF cut off for the GWAS , as genes with rare missense variants taken into functional testing, what was the MAF as I had thought 1% was used for the GWAS.

We used the following criteria for the GWAS: 1) all participating studies removed any variant with a minor allele count < 10; and 2) in our final meta-analysis, we removed all variants with a combined minor allele frequency < 0.001 and which were only present in three or fewer studies. See the following text in our methods section:

268 GWAS summary data were uploaded to central servers at the Broad Institute and the Digital Research Alliance of Canada and consortium-level quality control, including removal of variants with imputation quality ≤ 0.3 or minor allele count (MAC) < 10, was performed independently by two authors (A.M.S. and L.D.).

279 Variants with a minor allele frequency (MAF) of ≥ 0.001 and present in only one study or with a MAF < 0.001 and present in 3 or fewer studies were removed from the resulting meta-analysis summary files.

References

1. Marston NA, Garfinkel AC, Kamanu FK, Melloni GM, Roselli C, Jarolim P, Berg DD, Bhatt DL, Bonaca MP, Cannon CP, Giugliano RP, O'Donoghue ML, Raz I, Scirica BM, Braunwald E, Morrow DA, Ellinor PT, Lubitz SA, Sabatine MS, Ruff CT. A polygenic risk score predicts atrial fibrillation in cardiovascular disease. *Eur Heart J*. 2022. doi: 10.1093/eurheartj/ehac460
2. Aragam KG, Jiang T, Goel A, Kanoni S, Wolford BN, Atri DS, Weeks EM, Wang M, Hindy G, Zhou W, Grace C, Roselli C, Marston NA, Kamanu FK, Surakka I, Venegas LM, Sherliker P, Koyama S, Ishigaki K, Asvold BO, Brown MR, Brumpton B, de Vries PS, Giannakopoulou O, Giardoglou P, Gudbjartsson DF, Guldener U, Haider SMI, Helgadottir A, Ibrahim M, Kastrati A, Kessler T, Kyriakou T, Konopka T, Li L, Ma L, Meitinger T, Mucha S, Munz M, Murgia F, Nielsen JB, Nothen MM, Pang S, Reinberger T, Schnitzler G, Smedley D, Thorleifsson G, von Scheidt M, Ulirsch JC, Biobank J, Epic CVD, Arnar DO, Burt NP, Costanzo MC, Flannick J, Ito K, Jang DK, Kamatani Y, Khera AV, Komuro I, Kullo IJ, Lotta LA, Nelson CP, Roberts R, Thorgeirsson G, Thorsteinsdottir U, Webb TR, Baras A, Bjorkegren JLM, Boerwinkle E, Dedoussis G, Holm H, Hveem K, Melander O, Morrison AC, Orho-Melander M, Rallidis LS, Ruusalepp A, Sabatine MS, Stefansson K, Zalloua P, Ellinor PT, Farrall M, Danesh J, Ruff CT, Finucane HK, Hopewell JC, Clarke R, Gupta RM, Erdmann J, Samani NJ, Schunkert H, Watkins H, Willer CJ, Deloukas P, Kathiresan S, Butterworth AS, Consortium CAD. Discovery and systematic characterization of risk variants and genes for coronary artery disease in over a million participants. *Nat Genet*. 2022;54:1803-1815. doi: 10.1038/s41588-022-01233-6
3. Small AM, Melloni GEM, Kamanu FK, Bergmark BA, Bonaca MP, O'Donoghue ML, Giugliano RP, Scirica BM, Bhatt D, Antman EM, Raz I, Wiviott SD, Truong B, Wilson PWF, Cho K, O'Donnell CJ, Braunwald E, Lubitz SA, Ellinor P, Peloso GM, Ruff CT, Sabatine MS, Natarajan P, Marston NA. Novel Polygenic Risk Score and Established Clinical Risk Factors for Risk Estimation of Aortic Stenosis. *JAMA Cardiol*. 2024. doi: 10.1001/jamacardio.2024.0011
4. Boureau AS, Karakachoff M, Le Scouarnec S, Capoulade R, Cuffe C, de Decker L, Senage T, Verhoye JP, Baufreton C, Roussel JC, Dina C, Probst V, Schott JJ, Le Tourneau T. Heritability of aortic valve stenosis and bicuspid enrichment in families with aortic valve stenosis. *Int J Cardiol*. 2022;359:91-98. doi: 10.1016/j.ijcard.2022.04.022
5. Probst V, Le Scouarnec S, Legendre A, Jousseau V, Jaafar P, Nguyen JM, Chaventre A, Le Marec H, Schott JJ. Familial aggregation of calcific aortic valve stenosis in the western part of France. *Circulation*. 2006;113:856-860. doi: 10.1161/CIRCULATIONAHA.105.569467
6. Horne BD, Camp NJ, Muhlestein JB, Cannon-Albright LA. Evidence for a heritable component in death resulting from aortic and mitral valve diseases. *Circulation*. 2004;110:3143-3148. doi: 10.1161/01.CIR.0000147189.85636.C3
7. Small AM, Peloso G, Linefsky J, Aragam J, Galloway A, Tanukonda V, Wang L-C, Yu Z, Sunitha Selvaraj M, Farber-Eger EH, Baker MT, Setia-Verma S, Lee SSK, Preuss M, Ritchie M, Damrauer SM, Rader DJ, Wells QS, Loos R, Lubitz S, Thanassoulis G, Cho K, Wilson PWF, Natarajan P, O'Donnell CJ. Multiancestry Genome-Wide Association Study of

- Aortic Stenosis Identifies Multiple Novel Loci in the Million Veteran Program. *Circulation*. 2023. doi: 10.1161/circulationaha.122.061451
8. Chen HY, Dina C, Small AM, Shaffer CM, Levinson RT, Helgadottir A, Capoulade R, Munter HM, Martinsson A, Cairns BJ, Trudso LC, Hoekstra M, Burr HA, Marsh TW, Damrauer SM, Dufresne L, Le Scouarnec S, Messika-Zeitoun D, Ranatunga DK, Whitmer RA, Bonnefond A, Sveinbjornsson G, Danielsen R, Arnar DO, Thorgeirsson G, Thorsteinsdottir U, Gudbjartsson DF, Holm H, Ghouse J, Olesen MS, Christensen AH, Mikkelsen S, Jacobsen RL, Dowsett J, Pedersen OBV, Erikstrup C, Ostrowski SR, Regeneron Genetics C, O'Donnell CJ, Budoff MJ, Gudnason V, Post WS, Rotter JI, Lathrop M, Bundgaard H, Johansson B, Ljungberg J, Naslund U, Le Tourneau T, Smith JG, Wells QS, Soderberg S, Stefansson K, Schott JJ, Rader DJ, Clarke R, Engert JC, Thanassoulis G. Dyslipidemia, inflammation, calcification, and adiposity in aortic stenosis: a genome-wide study. *Eur Heart J*. 2023. doi: 10.1093/eurheartj/ehad142
 9. Verma A, Huffman JE, Rodriguez A, Conery M, Liu M, Ho YL, Kim Y, Heise DA, Guare L, Panickan VA, Garcon H, Linares F, Costa L, Goethert I, Tipton R, Honerlaw J, Davies L, Whitbourne S, Cohen J, Posner DC, Sangar R, Murray M, Wang X, Dochtermann DR, Devineni P, Shi Y, Nandi TN, Assimes TL, Brunette CA, Carroll RJ, Clifford R, Duvall S, Gelernter J, Hung A, Iyengar SK, Joseph J, Kember R, Kranzler H, Kripke CM, Levey D, Luoh SW, Merritt VC, Overstreet C, Deak JD, Grant SFA, Polimanti R, Roussos P, Shakt G, Sun YV, Tsao N, Venkatesh S, Voloudakis G, Justice A, Begoli E, Ramoni R, Tourassi G, Pyarajan S, Tsao P, O'Donnell CJ, Muralidhar S, Moser J, Casas JP, Bick AG, Zhou W, Cai T, Voight BF, Cho K, Gaziano JM, Madduri RK, Damrauer S, Liao KP. Diversity and scale: Genetic architecture of 2068 traits in the VA Million Veteran Program. *Science*. 2024;385:eadj1182. doi: 10.1126/science.adj1182
 10. Nazarzadeh M, Pinho-Gomes AC, Bidel Z, Dehghan A, Canoy D, Hassaine A, Ayala Solares JR, Salimi-Khorshidi G, Smith GD, Otto CM, Rahimi K. Plasma lipids and risk of aortic valve stenosis: a Mendelian randomization study. *Eur Heart J*. 2020;41:3913-3920. doi: 10.1093/eurheartj/ehaa070
 11. Thanassoulis G, Massaro JM, Cury R, Manders E, Benjamin EJ, Vasan RS, Cupple LA, Hoffmann U, O'Donnell CJ, Kathiresan S. Associations of long-term and early adult atherosclerosis risk factors with aortic and mitral valve calcium. *J Am Coll Cardiol*. 2010;55:2491-2498. doi: 10.1016/j.jacc.2010.03.019
 12. Cowell SJ. A Randomized Trial of Intensive Lipid-Lowering Therapy in Calcific Aortic Stenosis. *N Engl J Med*. 2005;352:2389-2397.
 13. Rossebø AB. Intensive Lipid Lowering with Simvastatin and Ezetimibe in Aortic Stenosis. *N Engl J Med*. 2008;359:1343-1356.
 14. Chan KL, Teo K, Dumesnil JG, Ni A, Tam J, Investigators A. Effect of Lipid lowering with rosuvastatin on progression of aortic stenosis: results of the aortic stenosis progression observation: measuring effects of rosuvastatin (ASTRONOMER) trial. *Circulation*. 2010;121:306-314. doi: 10.1161/CIRCULATIONAHA.109.900027
 15. International Schizophrenia C, Purcell SM, Wray NR, Stone JL, Visscher PM, O'Donovan MC, Sullivan PF, Sklar P. Common polygenic variation contributes to risk of schizophrenia and bipolar disorder. *Nature*. 2009;460:748-752. doi: 10.1038/nature08185

16. Chen EY. Enrichr: interactive and collaborative HTML5 gene list enrichment analysis tool. *BMC Bioinformatics*. 2013.

Reviewer Comments:

I have only one minor comment. Check for consistency in labelling in the paper transancestry and multi ancestry? Trans is mentioned in abstract.

We thank the reviewer for the additional comment distinguishing between the use of multi versus trans-ancestry. We removed all instances of trans-ancestry in the manuscript and replaced these with multi-ancestry.